# Spatial coalescent connectivity through multi-generation dispersal modelling predicts gene flow across marine phyla

Térence Legrand [1] ✉, Anne Chenuil [2], Enrico Ser-Giacomi [3], Sophie Arnaud-Haond[4], Nicolas Bierne [5] & Vincent Rossi [1] ✉

Gene flow governs the contemporary spatial structure and dynamic of populations as well as their long-term evolution. For species that disperse using atmospheric or oceanic flows, biophysical models allow predicting the migratory component of gene flow, which facilitates the interpretation of broad-scale spatial structure inferred from observed allele frequencies among populations. However, frequent mismatches between dispersal estimates and observed genetic diversity prevent an operational synthesis for eco-evolutionary projections. Here we use an extensive compilation of 58 population genetic studies of 47 phylogenetically divergent marine sedentary species over the Mediterranean basin to assess how genetic differentiation is predicted by Isolation-By-Distance, single-generation dispersal and multi-generation dispersal models. Unlike previous approaches, the latter unveil explicit parents-to-offspring links (filial connectivity) and implicit links among siblings from a common ancestor (coalescent connectivity). We find that almost 70 % of observed variance in genetic differentiation is explained by coalescent connectivity over multiple generations, significantly outperforming other models. Our results offer great promises to untangle the eco-evolutionary forces that shape sedentary population structure and to anticipate climate-driven redistributions, altogether improving spatial conservation planning.

Gene flow counterbalances natural selection and genetic drift, reshuffles mutations among spatial locations and contributes to shaping the contemporary spatial patterns of biodiversity[1–5]. By introducing foreign alleles to local populations, gene flow spreads adaptative changes and tends to alleviate the effect of inbreeding depression[1–3]. Simultaneously, gene flow homogenizes allele frequency among populations, which counteracts the effects of local adaptation, reducing the mean fitness of populations (i.e., migration load)[2]. Fundamentally, gene flow is ensured when dispersal is followed by reproduction and subsequent offspring survival[6]. A common confusion prevails between demographic connectivity (i.e., the number of migrants exchanged among populations), which is usually assessed by direct detection of individuals (field observations and parentage analyses of genetic data), and genetic connectivity (i.e., the efficient transfer of genetic material between distant populations), which is indirectly estimated thanks to population genetics analyses[1,3,4,7–10]. In

[1]Aix Marseille University, Universite de Toulon, CNRS, IRD, Mediterranean Institute of Oceanography (UMR 7294), Marseille, France. [2]IMBE, CNRS UMR 7263, Aix Marseille Université, Avignon Université, IRD 237, Station marine d'Endoume, Chemin de la Batterie des Lions, 13007 Marseille, France. [3]Department of Earth, Atmospheric and Planetary Sciences, Massachusetts Institute of Technology, 54-1514 MIT, Cambridge, MA, USA. [4]MARBEC (Marine Biodiversity, Exploitation and Conservation, UMR 9190) Univ. Montpellier, IFREMER, IRD, CNRS, Sète, France. [5]ISEM, Univ Montpellier, CNRS, IRD, Montpellier, France. ✉e-mail: legrandterence@gmail.com; vincent.rossi@mio.osupytheas.fr

this way, demographic and genetic connectivity seem to interact on specific -yet poorly appreciated- temporal and spatial scales[3,11–14]. This may explain the numerous mismatches reported between demographic connectivity and gene flow estimates[7–9].

This paradox could stem from the fact that dispersal is a complex and multi-aspect process involving interlocked ecological and evolutionary mechanisms[6,15,16]. Dispersal results from movements of individuals themselves or movements induced by third parties categorized as biotic (e.g., thanks to other moving organisms[17,18]) or abiotic, that is driven by winds and ocean currents[8,16,19]. This study focuses on species that, as adults, have no or little displacement abilities (hereafter called sedentary species) so that their connectivity is mostly ensured by the abiotic dispersal of propagules, like numerous marine and terrestrial taxa. This alleviates the difficulties in appraising the movements of wild populations and biotic third parties. In marine sedentary populations, early-life non-motile stages (e.g., seeds, eggs, larvae) are regularly released in the water column and are then passively transported across the seascape by anisotropic currents over various spatial scales[20,21], ensuring the replenishment of both local and distant populations[22,23]. As such, the proper evaluation of current-driven dispersal should help us disentangling the evolutionary forces (gene flow, selection, genetic drift or mutation) shaping marine biodiversity and its climate-driven redistributions[24]. The inherent spatial scales of genetic structures are generally a few orders of magnitude higher than potential dispersal distances over a single generation[25], even for species exhibiting rare long-distance dispersal[26,27]. Likewise, a single-generation dispersal event does not allow to evaluate the evolutionary timescale over which gene flow shapes genetic diversity[6]. Theory predicts instead that consecutive dispersal events of numerous propagules, acting in synergy with other evolutionary forces, shape together the genetic diversity observed at broad-scale[26,28]. Consequently, modelling genetic diversity from the unique perspective of water-borne dispersal should enlighten the typical scales and relative importance of evolutionary forces that shape the spatial structure of marine sedentary populations.

Modelling water-borne dispersal is a multidisciplinary challenge sharing tight commonalities with air-borne dispersal[29]. First, it requires to jointly account for the spatio-temporal variability of currents, the species-specific early-life traits, the habitat patchiness[9,30], as well as to consider all possible connectivity pathways[31]. Second, it must simulate consecutive dispersal events by considering multiple generations of migrants where each intermediary connected population acts as a stepping-stone. Third, it must simulate all possibly existing populations, not just known or sampled ones. Biophysical models, which simulate explicitly the dispersal of propagules by oceanic chaotic flows, have been widely used in the last few decades to derive physical connectivity metrics such as dispersal kernels[20,23,32,33]. Simulations of single-generation dispersal commonly provide quantitative estimates of how distant populations are connected with each other. However, they rarely match observed gene flows[8], possibly due to intrinsic flaws such as disregarding the multi-generational character of successive dispersal events[34–36], while overlooking intermediate stepping-stone connections.

In the marine realm, current models considering multi-generational dispersal are seldom used, concern only a few specific species or taxa, and still inadequately explain genetic differentiation measures. They rely on the computation of connectivity matrices, which are mathematical objects describing dispersal of propagules exchanged between discrete populations, hereafter called localities. Such matrices can be interpreted as adjacency matrices of directed and weighted networks (or graphs). Thus, an approach consists in considering network theory tools such as shortest paths analysis to estimate the strength of connections among two distinct localities over multiple dispersal events[34,37,38]. As shortest and most-probable paths of such networks differ[39], these

methods neglect all other possible pathways that may drastically change the resulting connectivity[40]. Another approach uses Markov chains and matrix multiplications to estimate the probability of connection between locality-pairs over a given number of generations[35,41–43]. Studies using this method did not consider all the inherent dispersal pathways as they only assessed the connection probabilities occurring at a given number of generations (equivalent to the exact number of multiplication) while neglecting all intermediate connections associated to any number of generations lower or equal to the prescribed number of dispersal events[44]. Moreover, and to our knowledge, all present modelling approaches simulate stepping-stone dispersal of single lineages; in other words, they estimate the connectivity resulting from explicit parents-to-offspring connections, i.e., filial connectivity. However, it is conspicuous that two fully disconnected localities, which are both replenished by migrants originating from the same source locality, should share common alleles and thus display similar allele frequencies.

The consideration in dispersal models of this conceptual view of coalescent connectivity[44], which highlights implicit links among siblings through common ancestors has been overlooked to-date, although it could largely alter gene flow predictions and contribute to the aforementioned discrepancies between predicted dispersal and realized gene flow assessments[1]. Note that here the term "coalescent" refers to the dispersal model, not to the genetic model. Modelling such implicit connections would permit going beyond the concept of pairwise interactions addressing higher-order dynamics, a perspective that is lately attracting lots of interests, both in Network Theory[45] and Theoretical Ecology[46,47].

In this work, we present an exhaustive comparison between demographic and genetic basin-scale connectivity based on classical and novel dispersal metrics across a meta-analysis of several marine taxa. While our results mainly apply to marine sedentary populations whose dispersal is mediated by ocean currents, our models and conclusions have the potential to transform how water- as well as air-borne dispersal of sedentary terrestrial populations are evaluated. Here, we introduce state-of-the-art multi-generation dispersal models that evaluate all connections among locality-pairs for a fixed number of generations while simultaneously accounting for those accumulated by previous generations[44]. Our models provide not only a precise estimation of the explicit links (filial connectivity) but also allow computing, for the first time, the implicit links existing between any locality-pair having common source localities through multi-generational dispersal (coalescent connectivity). After parameterizing our model with the main dispersal traits of various taxa encompassing seagrasses, algae and metazoans, we test modelled gene flow predictions against an extensive compilation of observed genetic structures (i.e., genetic differentiation estimates given by $F_{st}$ between locality-pairs[48]) over the whole Mediterranean Sea. We find that coalescent connectivity through multi-generation dispersal explains almost 70% of the observed variance of genetic structures, substantially improving gene flow predictions with respect to previous approaches. Furthermore, the optimal number of generations to best predict gene flow significantly correlates with the sampling coverage scaled by the species-specific dispersal abilities, enlightening the typical scales of eco-evolutionary processes. It suggests that our model could be used to infer population genetic structures, a key prerequisite for management and protection.

## Results

We test the predictions of our multi-generation explicit and implicit dispersal models[44], simulating filial and coalescent connectivity respectively, against an extensive compilation of 58 genetic structures observed in the Mediterranean Sea. The dataset contains 3821 observed $F_{st}$ measures ($F_{st}^{obs}$) between locality-pairs for 47 marine

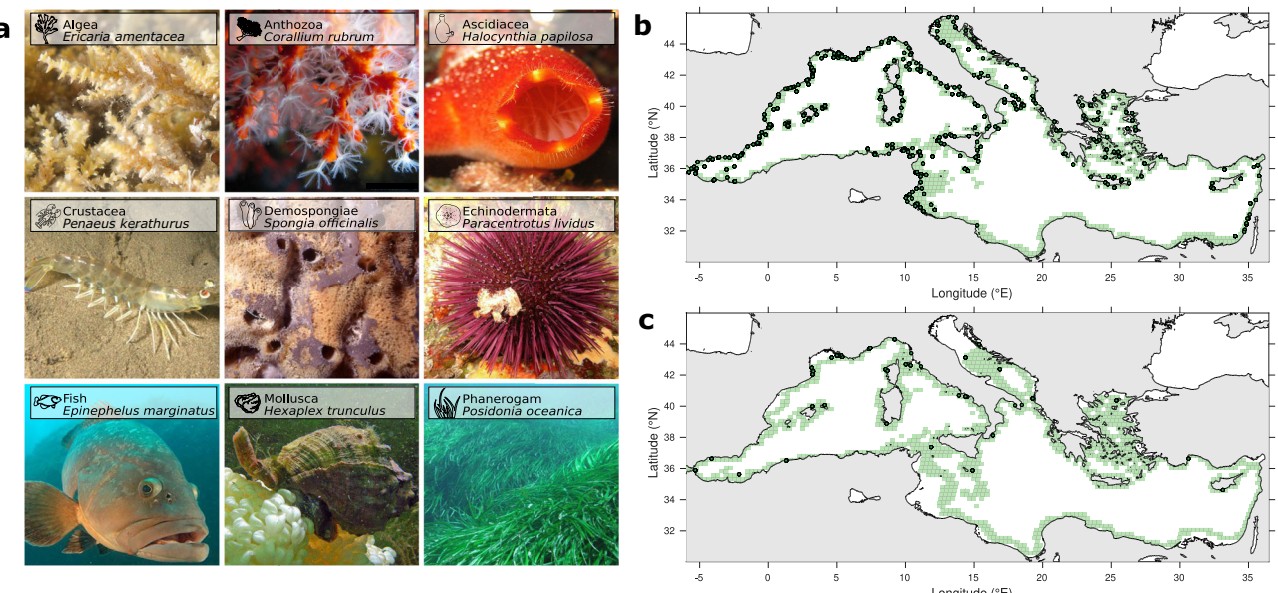

**Fig. 1 | Meta-analysis summary, estimated habitats and geographical locations of the sampled populations. a** Exemplary pictures of marine species belonging to the nine taxonomic groups comprised in the compilation of 58 population genetics studies. **b** Basin-scale view of all sampled populations compiled in the meta-analysis (dark green dots) and putative localities (light green squares) that act as stepping-stones in our multi-generation dispersal model for the shallow coastal habitat.

**c**, same as **b** but for the neritic shelf habitat. Source data are provided as a Source Data file. Photos credit (from left to right, top to bottom): © Veronique Lamare 2015, © Alain-Pierre Sittler 2004, © Alain-Pierre Sittler 2005, © Gilles Cavignaux 2007, © Jean-Georges Harmelin 1999, © Frédéric André 2007, © Christophe Dehondt 2006, © Jean-Claude Wolles 2007 and © Jean-Georges Harmelin 2005 (published on the DORIS web site, https://doris.ffessm.fr/).

species (Fig. 1a) which are characterized by a biphasic life cycle, i.e. early-life free-swimming dispersing propagules and full to semi-sedentary adult (Fig. 2a). We model the full range of variability of current-driven dispersal over the whole Mediterranean basin for each species using a fine-tuned particle-tracking model[23,49–52], fed by the horizontal multi-year velocity field from an operational data-assimilative ocean model[53]. Each species is characterized by three main dispersal traits: Pelagic Larval Duration (PLD, i.e. the time propagules spend drifting with ocean currents, Fig. 2a), spawning seasons and adult habitats (Fig. 1b, c). These biological factors were identified as the major ones governing how the variability of ocean currents reflect on our probabilistic connectivity metrics[39,50].

Among the 58 compiled basin-scale population genetic studies (Fig. 1), we test for prediction of $F_{st}^{obs}/(1 - F_{st}^{obs})$[48,54] by our cumulative explicit and implicit dispersal models (Fig. 2b, c) considering both single- and multi-generation estimates thanks to a maximum-likelihood population-effects (MLPE) linear mixed model. We also test for conventional Isolation-By-Distance (IBD) models, using either Euclidian or sea-least-cost distances (i.e., the shortest overwater distances). The number of significant $R^2$ is 16 for single-generation explicit, 30 for Euclidian IBD, 31 for sea-least-cost IBD, 37 for multi-generation explicit and implicit models respectively (Fig. 3). Among the significant MLPE linear mixed model $R^2$, the lowest mean $R^2$ is found for the single-generation explicit dispersal model (0.50), followed by both sea-least cost (0.58) and Euclidian (0.59) IBD models. The highest mean $R^2$ stem from both multi-generation models, with 0.68 for explicit dispersal and 0.69 for implicit dispersal models.

Next, we evaluate the quality (i.e., predictive ability) of each model relative to each of the other models, and then to all models together, thanks to the AIC estimator. To do so, the same expected value, $F_{st}^{obs}/(1 - F_{st}^{obs})$, is considered among the five different models of gene flow when using MLPE linear mixed models. The best model is the multi-generation implicit dispersal model in 25 cases, the multi-generation explicit dispersal model in 18 cases, the Euclidian IBD model in 10 cases, the sea-least-cost IBD model in 7 cases, and the single-generation explicit dispersal model in only 3 cases. The relative

likelihood permits to evaluate that among the five candidate models, the multi-generation implicit dispersal model is the one that minimizes the information lost with a mean relative likelihood of 0.75 at the meta-analysis scale. Moreover, this model is the only one that displays positive relative likelihood difference in pairwise comparisons with the four other models (Fig. 3). Hence, our multi-generation implicit dispersal model provides the best predictions of $F_{st}^{obs}$.

When inspecting the study-specific accuracy of the best multi-generation implicit dispersal model, $R^2$ values range from 0.11 for the colonial ascidian *Botryllus schlosseri*[55] to 1 for the ascidian *Halocynthia papillosa*[56], the sea cucumber *Holothuria mammata*[57] and the sea snail *Phorcus turbinatus*[56] (Fig. 4a). For studies that include abundant genetic markers, it is possible to identify markers with particularly high $F_{st}^{obs}$ values (i.e., outliers; based on appropriate models), suggesting that natural selection filtered alleles differentially among localities. For a sea urchin *Paracentrotus lividus*[58], $F_{st}^{obs}$ outlier loci returns a $R^2$ of 0.93***, which is higher than the $R^2$ of 0.80*** obtained considering all the loci. Note that two studies focusing on the same species both using microsatellite markers, can display contradictory results: *Corallium rubrum* is characterized by a highly significant and tight correlation ($R^2 = 0.54$***, ref. 59) as well as a non-significant loose correlation ($R^2 = 0.27$ns, ref. 60; Fig. 4a), exemplifying inter-study variability. For the flathead grey mullet *Mugil cephalus*[61], which have been sampled homogeneously across the Mediterranean basin (i.e., the mean straight-line geographical distance between sampled localities, $D_{btw}$, is 1279 km, Supplementary Table 1), the network representation of modelled $F_{st}$ ($F_{st}^{mod}$) mimics well the one of $F_{st}^{obs}$ (Fig. 4b). $F_{st}^{obs}$ are low for locality-pairs located in the western basin but relatively high between western and eastern Mediterranean localities, suggesting spatial genetic structuring that is well predicted by coalescent connectivity (see the scatter plot of Fig. 4b, $R^2 = 0.73$***). Similar results are obtained for instance for a seagrass species, *Cymodocea nodosa*[62] (Fig. 4c). Although in this case spatial sampling is less balanced (sampled locality mostly clustered along the Spanish coast but with two distant localities in the Eastern Mediterranean, $D_{btw} = 1601$ km), the

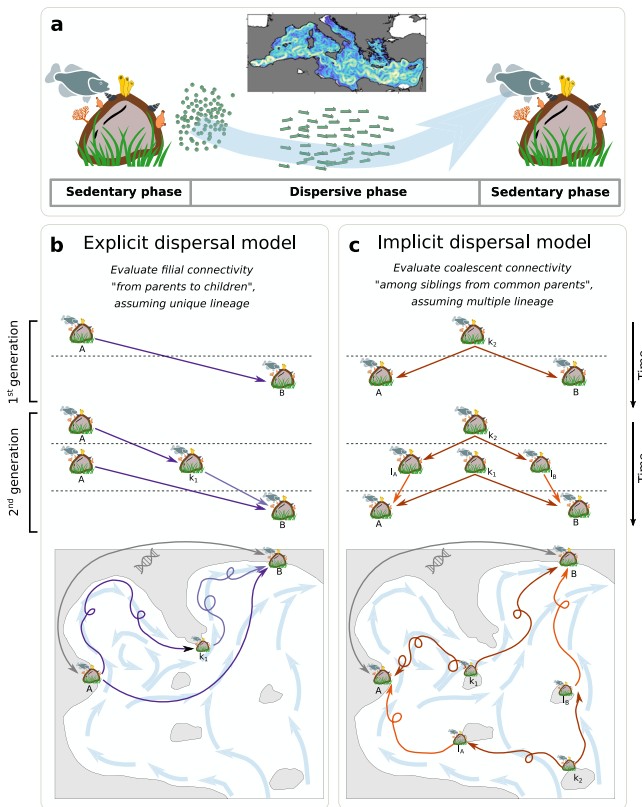

**Fig. 2 | Modelling multi-generational water-borne dispersal. a** Schematic representation of a biphasic life cycle composed of sedentary adults and dispersive early-life stages, that is a distinctive feature of all species included in the meta-analysis. During the dispersive phase, numerous individuals are dispersed across the seascape by turbulent currents (represented in the Mediterranean miniature). **b** Explicit dispersal model evaluates filial connections between locality-pairs. **c** Implicit dispersal model estimates coalescent connections between locality-pairs. Schematics highlight the simulated genealogy over two generations and illustrate one of many multi-generational dispersal pathways that are considered by our models when estimating the connectivity between distant localities A and B. Icons credit: © vectors market, © Agne Alesiute, © Elisabetta Calabritto, © Luis Prado, © Joi Stack, © Tatina Vazest, © mindgraphy, © Oleksandr Panasovskyi, © ProSymbols, © Sean Maldjian (changes were made on all the icons, Creative Commons BY 4.0 license).

genetic structure (high $F_{st}^{obs}$) between the Adriatic and Spanish localities is well reproduced by $F_{st}^{mod}$ ($R^2 = 0.77$***).

We then test the robustness of the multi-generation implicit dispersal model with respect to the species and studies attributes. None of these factors (taxa, PLD, spawning season, genetic marker, and $D_{btw}$) has a significant effect on model fit results ($R^2$, $p$ value, Supplementary Table 10). Yet, we find a significant linear negative correlation between the logarithm of MLPE linear mixed models *p-values* and the number of sampled localities ($R^2 = 0.40$***) as well as with the range of $F_{st}^{obs}$ ($R^2 = 0.18$***, Supplementary Table 10). Furthermore, the probability to obtain a significant $R^2$, i.e., successful gene flow predictions, as a function of categories of number of sampled localities is well predicted by a logit model using a binomial distribution ($R^2 = 0.72$***, Supplementary Fig. 10). Finally, we find a positive linear relationship between the PLDs categories and $D_{btw}$ divided by the optimal number of modelled generations ($R^2 = 0.31$***, Fig. 5), considering the 37 studies whose model predictions of genetic observations are significant. This surprisingly tight relationship has several interesting implications. First, if the spatial structures of two species are evaluated through the same sampling design, our dispersal model needs more generations for short PLD than for long PLD species to properly represent genetic

observations. Consequently, our model conforms to the intuitive view that a species needs more successive events of dispersion across generations to disperse widely across the seascape. Moreover, if two species have similar PLDs, our model requires a higher number of generations to accurately simulate the genetic structure of the one whose sampling is wider and more comprehensive. Last, when parameterized to fit the dispersal ability (i.e., PLD) of the target species, the predictive ability of the implicit dispersal model scales with the $D_{btw}$. The metric $D_{btw}$/(optimal number of generation) could be interpreted as the average velocity of genetic connectivity in km.generations⁻¹. Altogether and assuming that the species-specific dispersal traits have been accurately parametrized, it suggests that the non-significant gene flow predictions (e.g., 21 studies among the meta-analysis) could be attributed to too scarcely and spatially-restricted sampling rather than to the models abilities themselves.

## Discussion

For gene flow predictions at the meta-analysis scale, the cumulative multi-generation implicit dispersal model[44], which evaluates coalescent connectivity, significantly outperformed explicit dispersal models, which assess filial connectivity, as well as IBD models. Nearly two-thirds of the predicted genetic differentiation estimates are significant, even though observations span a wide phylogenetic range of sedentary taxa with contrasted dispersal traits. It is more than twice the proportion displayed by the explicit single-generation dispersal model, which emerges as the worst model in our meta-analysis. Overall, the best models are the multi-generation implicit and explicit dispersal, suggesting unambiguously that modelling multiple generations is crucial to accurately predict genetic connectivity[36]. To our knowledge, the only other model that considers both coalescent and filial connectivity uses circuit theory to approximate how barriers and corridors of habitat affect genetic connectivity through a process called Isolation-By-Resistance[63]. While this empirical model helped interpreting gene flow for self-dispersing organisms across a well-known and relatively stable landscape[40], it has not yet been applied to the marine realm probably because the seascape is highly variable and in perpetual movement. Contrarily, our dispersal models are mechanistic and plainly consider the dynamical properties of ocean currents that drive water-borne dispersal, so that it can be readily applied to air-borne dispersal[29].

We find that multi-generation dispersal models performed significantly better than IBD models. Similar results are found when using explicit multi-generation models for the seagrass *Zostera marina* in the North Sea[35], and the mollusc *Kelletia kelletii* along the Californian shores[43]. Yet, Boulanger et al.[34] found a tighter and more significant correlation of observed genetic structure with sea-least-cost IBD model than with their explicit multi-generation dispersal model for the fish *Mullus surmuletus*[64]. By contrast, when using the same data of $F_{st}^{obs}$, our multi-generation implicit dispersal model returns a better correlation than IBD models.

Our results also show that IBD models (Euclidian and sea-least-cost distance) better explain observed genetic differentiation than the single-generation dispersal model, in accord with previous studies[34,35]. Slightly more than half of the compiled studies displayed significant IBD predictions. The mean Mantel $R^2$ computed from these studies (Supplementary Table 5 and 6) is comparable to previous meta-analysis[65,66]. Still, for some previous studies which do not consider multi-generation, single-generation dispersal models were reported to improve gene flow prediction as compared to IBD models (e.g., refs. [67–69]). The apparent contradiction with the present results may be due to a publication bias: single-generation dispersal predictions that were worse than IBD's ones could have been withheld by authors. Single-generation dispersal models are worse than IBD models probably because broad-scale single-generation dispersal modelling studies often reported that most distant localities were not connected at

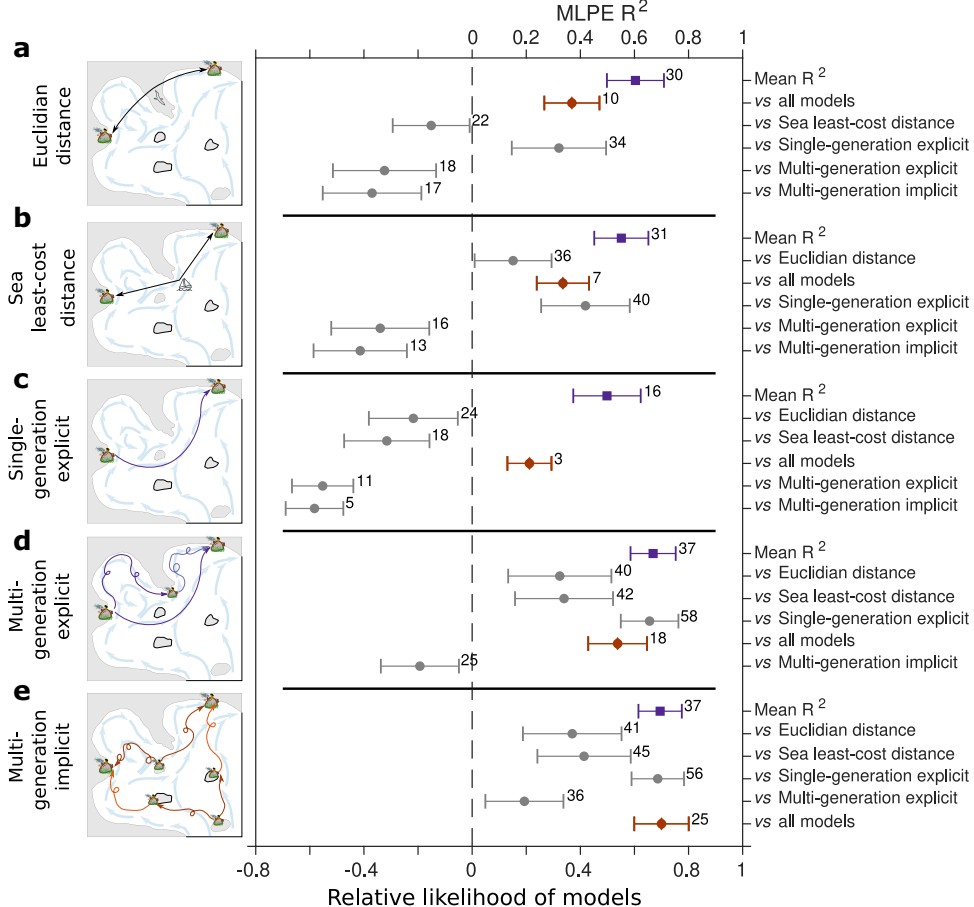

**Fig. 3 | Cross-comparison of gene flow predictive models.** Purple squares indicate mean MLPE linear mixed model $R^2$, red diamonds indicate the relative likelihood of the reference model (depicted on the left) vs the four remaining models, and grey dots indicate the relative likelihood difference of the reference model (depicted on the left) vs each of the four remaining models. Comparative analyses are made for **a** IBD (Euclidian) model, **b** IBD (sea-least-cost) model, **c** single-generation explicit dispersal model, **d** multi-generation explicit dispersal model and **e** multi-generation implicit dispersal model. Only the significant predictions (*p* value*) of each reference model (left) are considered to compute the mean $R^2$, while all the 58 studies are used to compute the relative likelihood and the relative likelihood difference. The number of significant predictions per model (over the total of 58 studies) is reported on the right of each purple square. The number of times the reference model displays the lowest AIC among all the four other models is reported on the right of each red diamond. The number of times the reference model displays the lower AIC among each of the four remaining models is reported on the right of each grey dots. Error-bars represent the 95 % confidence intervals. Source data are provided as a Source Data file. Icons credit: © vectors market, © Agne Alesiute, © Elisabetta Calabritto, © Luis Prado, © Joi Stack, © Tatina Vazest, © mindgraphy, © Sean Maldjian (changes were made on all the icons, Creative Commons BY 4.0 license).

all, suggesting genetic isolation. This was often explained by dispersal barriers due to major oceanographic features such as fronts and jet-like currents[70]. Our results contradict this view: multi-generation dispersal models suggest that these locality-pairs can be connected through stepping-stone dispersal despite the supposed physical barriers[44]. Notwithstanding, our models showed that physical barriers still can limit the levels of connectivity (see Fig. 8 of Ser-Giacomi et al.[44]) as reflected in observed genetic differentiation and as predicted by theoretical gene flow magnitudes[3,7]. Consequently, the reduced gene flow measured across these barriers is probably due to the large environmental gradients (affecting negatively propagules survival and settlement) rather than to intrinsic transport barriers (as propagules effectively disperse through them[44]). Since IBD is an analytical model with little biological meaning as it tentatively explains genetic differentiation by only accounting for geographical distances, it does not allow disentangling the relative importance of evolutionary processes that control gene flow. In line with the distinction between IBD and Isolation-by-Environment[71], and since our mechanistic multi-generation dispersal models realistically simulate stepping-stone dispersal, one can tease apart the respective role of evolutionary forces in driving gene flow. In fact, the non-significant predictions of observed

genetic differentiation suggests that strong evolutionary forces (such as natural selection) not considered in our approach are at play. Moreover, our analysis indicates that addressing implicit connections and thus going beyond simple pairwise perspectives (i.e., such as explicit connections)[45], can significantly improve the understanding of biogeographical and genetic patterns[47]. Altogether, our results suggest that the supposed physical barriers often underlined in seascape analyses are more permeable to dispersal than previously thought[70], and that genetic isolation in the marine realm could be rather due to environmental selection acting on drifting propagules and settled adults as well as intrinsic reproductive isolation[72]. Since ocean currents[73], transport and mixing processes[74], as well as ocean temperatures[75], are already changing fast, the structure of marine populations is expected to fluctuate accordingly, consistently with the recent evidence of spatial reorganization of marine biodiversity[24].

Our models perform better than previous ones probably also because they consider properly the mesoscale variability of ocean currents, they are parametrized with species-specific dispersal traits, and they allow testing explicitly what number of generations maximize correlations with observed data. Since there is no consensus on the adequate number of generations required to

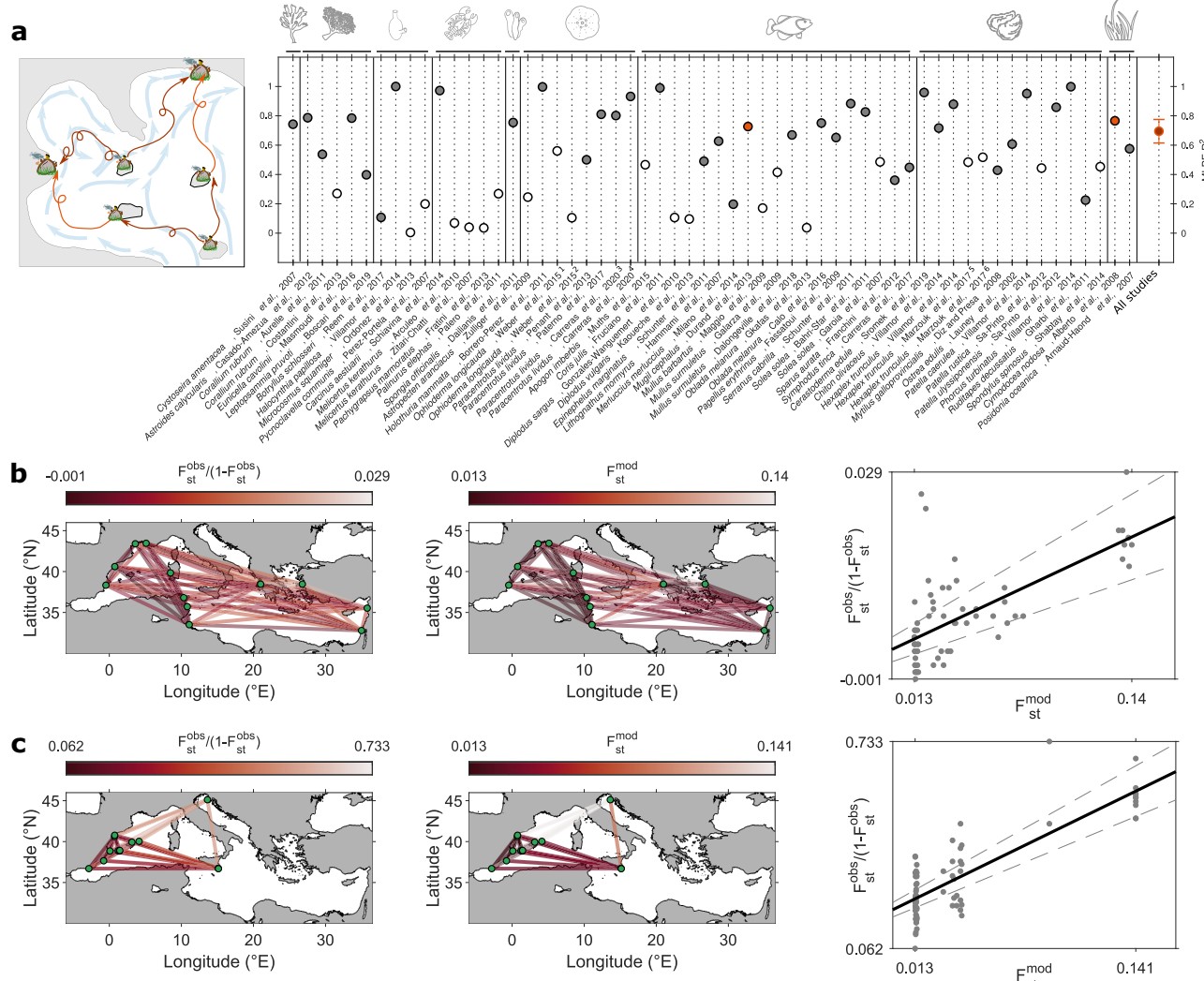

**Fig. 4 | Accuracy of the multi-generation implicit dispersal model in explaining compiled genetic structures. a** MLPE linear mixed model $R^2$ between $F_{st}^{mod}$ and $F_{st}^{obs}/(1 - F_{st}^{obs})$. Filled dots highlight the 37 significant correlations (*p* value*). Note that some results reported by a given study are analyzed separately: (i) Weber et al.[87] used nuclear DNA marker ([1]) and mtDNA marker ([2]); (ii) Carreras et al.[58] considered all the loci together ([3]) and then only the Mediterranean outliers loci ([4]); (iii) Marzouk et al.[88] analyzed nuclear DNA marker ([5]) and mtDNA marker ([6]). Error-bars for all studies represent the 95 % confidence intervals. **b** Network representation of $F_{st}^{obs}/(1 - F_{st}^{obs})$ (observed $F_{st}$, left) and $F_{st}^{mod}$ (modelled $F_{st}$, right) and their corresponding scatterplot for the flathead grey mullet (*Mugil cephalus*;[61] red dot in a). **c** same as **b** but for a seagrass (*Cymodocea nodosa*;[62] red dot in **a**). Source data are provided as a Source Data file. Icons credit: © vectors market, © Agne Alesiute, © Elisabetta Calabritto, © Luis Prado, © Joi Stack, © Tatina Vazest, © mindgraphy, © Oleksandr Panasovskyi, © ProSymbols, © Sean Maldjian (changes were made on all the icons, Creative Commons BY 4.0 license).

comprehend gene flow, previous multi-generational approaches used shortest path algorithms (minimum number of steps to connect sampled locality-pairs) with 25 or less intermediate steps[34,37] or set arbitrarily the number of generations from dozens (as it was considered sufficient to span the studied domain[35]) to thousands (i.e., the number of Markov chain iterations needed to reach convergence) of generations[43]. It illustrates that the typical time scales over which demographic connectivity interplays with genetic connectivity are difficult to infer[1,3]. The relationship between the main dispersal trait (PLD) and the spatial extent of the sampling divided by the optimal number of generations ($D_{btw}$/Opt gen, see Fig. 5) implies that the temporal scales estimated with our implicit model (from 1 to several tens of generations, e.g., ecological time) and spatial scales derived from genetic methods (over typical evolutionary scales, from a hundred to a few thousands of km) are tightly linked. This is aligned with estimates of dispersal kernels that were found congruent over ecological and evolutionary time[14]. Indeed, the averaged genetic connectivity velocities, in km.generation$^{-1}$,

match the demographic connectivity scales for similar PLD (an order of 100 km for two coastal fishes[23]). Moreover, island model theory assumes that the metapopulation has reached an equilibrium between gene flow and genetic drift[76], suggesting that the implicit model should predict best $F_{st}^{obs}/(1 - F_{st}^{obs})$ for long-term multi-generation dispersal, that is when dispersal probabilities reach convergence[43] (i.e., after about 500 generations in our case; see Fig. 9 of Ser-Giacomi et al.[44]). The relatively low median optimal number of generations disclosed here (~20, Supplementary Fig. 7) further suggests that gene flow and drift have insufficient time to reach equilibrium due to environmental stochasticity and rapidly changing ecological forces[10]. Moreover, the substantial impacts of ecological processes on genetic structures shown here could explain why chaotic genetic patchiness has been recently documented at small space and time scales[11,77–81], which are also the scales over which dispersal and environment co-vary. As such, dispersal could be characterized as one of the evolutionary forces shaping the contemporary spatial patterns of biodiversity, along with natural

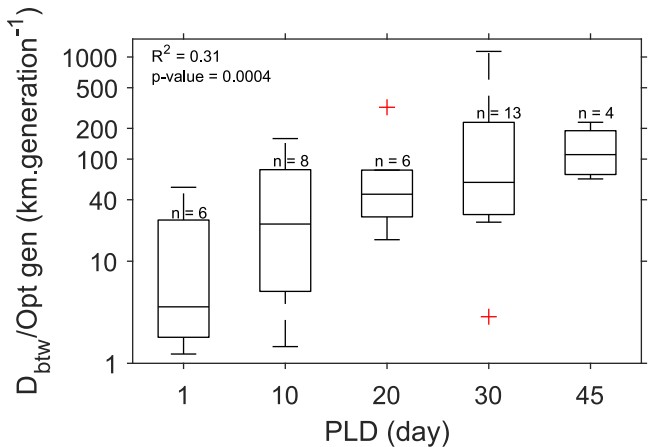

**Fig. 5 | Sensitivity of the multi-generation implicit dispersal model predictions to dispersal trait and sampling design.** Correlation ($R^2 = 0.31***$ using a linear regression model fit) between the PLDs categories (x-axis) and the proxy of sampling extent $D_{btw}$ divided by the optimal number of modelled generations (y-axis). $n$ = number of studies whose model predictions of genetic observations are significant per PLD categories. On each box, the central mark, the bottom and top edges indicate the indicates the median, the 25th and the 75th percentiles, respectively. The whiskers extend to the most extreme data points not considered outliers, and the outliers are plotted individually using a cross symbol. Note that Susini et al.[91] has been removed from this figure as it displays the minimal number of localities of our compilation (i.e., three localities while two sampled populations are indeed compassed in the same locality), which does not permit a robust evaluation of $D_{btw}$/Opt gen ratio. When considering this study, $R^2 = 0.23***$. Source data are provided as a Source Data file.

selection, providing evolutionary changes occurring over ecological time scales of few generations[78,82].

Last but not least, our multi-generation mechanistic dispersal models, which allow assessing both filial and coalescent connectivity and are applicable to other taxonomic groups and ecosystems, could serve as future guidelines to optimize sampling design for population genetic studies and anticipate the structure of wild sedentary populations. Moreover, while higher-order interactions can stabilize competitive ecosystems[83], they can also foster abrupt transitions[45]. Investigating such aspects in marine ecosystems would provide crucial information about how they could react to future perturbations and open the way to novel approaches to spatial ecology. In the context of biodiversity loss[84] and spatial reorganization[24], it is indeed urgent to better understand the eco-evolutionary dynamics that continuously shape population structures to improve protection strategies[85] and management of natural resources such as forestry and fisheries[86].

## Methods

### Studies characteristics

We screen the published literature over the last two decades to gather population genetic studies focussing on marine species at basin-scale in the Mediterranean Sea. While our meta-analysis intends to be the most comprehensive possible in terms of collected data and taxa covered, pre-selected studies are filtered out based on two criteria: the biological traits of the species and the sampling design. In this way, we exclude datasets that are not appropriate to address our research question while optimizing statistical discriminatory power. More specifically, we select studies (i) that target poorly mobile species, i.e., characterized by a biphasic life cycle with early-life free-swimming dispersing propagules (e.g., seeds, eggs, larvae or body fragments) and sessile to sedentary adult behaviour (Fig. 1a) and (ii) which must present at least four distinct localities where at least 15 individuals were sampled (see the PRISMA flow diagram depicting our systematic review in Supplementary Notes 1). As a result, this meta-analysis

compiles 58 population genetic studies published between 2002 and 2020, encompassing 47 different marine species distributed in nine taxonomic groups: Algae, Anthozoa, Ascidiacea, Crustacea, Demospongiae, Echinodermata, Fish, Mollusca and Phanerogam (Fig. 1a). In total, 559 localities were sampled across the basin (dark green dots in Fig. 1b, c), representing 3821 locality-pairs. $F_{st}$ fixation index allows evaluating the genetic differentiation (i.e., observed genetic differentiation, $F_{st}^{obs}$) between locality-pairs using five types of genetic markers (allozymes, microsatellites, mitochondrial DNA sequences, nuclear DNA sequences and SNPs from high throughput sequencing). Note that, when applicable, we separately analyze different genetic markers extracted from the same study, i.e., mitochondrial DNA sequences and nuclear DNA sequences for *Ophioderma longicauda*[87] and *Hexaplex trunculus*[88].

To gauge the sampling strategy of each selected study, we compute the mean straight-line geographical distance (in km) between sampled localities. This geometrical metric, that we called $D_{btw}$, quantitatively evaluates the spatial coverage of the sampling strategy carried out in each study.

### Species characteristics

Based on literature review (Supplementary Table 2), all species are classified according to their main dispersal traits. Reproductive phenology comprises five groups to reflect seasonal (spring, summer, fall, winter) and annual spawning strategies. Pelagic Larval Durations (PLD) are categorized in five groups: very-low (1 day), low (10 days), low-to-medium (20 days), medium-to-high (30 days) and high (45 days) dispersal abilities. Finally, two broad-scale classes of habitats are distinguished (Supplementary Methods 2): the shallow coastal habitat (inner continental shelf whose depths span 0–50 m; Fig. 1b) and the neritic shelf habitat (mid to outer continental shelf whose depths range is 50–200 m; Fig. 1c).

### Biophysical modelling

Tracking passive Lagrangian particles is a common approach to characterize flow-driven dispersal of propagules[20,23,29,49,51,89] (Fig. 2a). To provide synthetic -yet realistic- views of basin-scale propagules dispersal while encompassing the full variability of ocean currents and for various dispersal abilities, we use the Lagrangian Flow Network framework (LFN[23,39,49–52]). It combines a particle-tracking model with graph theory tools to generate and analyse connectivity matrices (Supplementary Methods 3), allowing us to investigate oceanic dispersal in a robust and efficient manner[50].

The Mediterranean Sea surface characterized by favourable habitats is subdivided into several ¼° sub-areas that are considered in our analysis as theoretically isolated marine localities, resulting in $n = 1170$ localities in the shallow coastal habitat (Fig. 1b) while the neritic shelf habitat is composed of $n = 1163$ localities (Fig. 1c; see Supplementary Methods 2). For each LFN experiment, we track 100 propagules per localities (i.e., totalling ~120,000 propagules considering all the localities contained in each habitat) by integrating daily gridded velocity fields generated by a data-assimilative operational ocean model implemented in the Mediterranean Sea at a 1/16° horizontal resolution[53]. We use the horizontal flow field at 10 m and 100 m for species inhabiting shallow coastal and neritic shelf habitats, respectively. Overall, virtual propagules trajectories are modelled at two specific depths during the five PLDs groups defined previously, simulating consecutive propagule release events with a 10-day periodicity over 2000–2010. Assuming that the long-term (e.g., decadal, centennial and longer) variability of ocean currents is negligible as compared to their inter- and intra-annual variations, our approximation using 10 recent years allows comprehending the full variability of both contemporary and past oceanic flows. By computing billions of Lagrangian trajectories and recording their initial and final positions, the LFN constructs 402 connectivity matrices for each of the 10

habitat/PLD combinations, resulting in 4020 matrices in total. The elements $m_{ij}$ of each raw $n \times n$ connectivity matrix encode the number of propagules advected between all locality-pairs; they are converted into backward-in-time dispersal probabilities thanks to a column-normalization:

$$m_{ij} = \frac{m_{ij}}{\sum_{i=1}^{n} m_{ij}} \qquad (1)$$

Then connectivity matrices $M$ are aggregated (i.e., element-by-element averaged) according to their starting dates to match each species-specific spawning phenology. We average 402 matrices for year-round release and about 100 matrices for seasonal release (i.e., 402 and about 100 Lagrangian experiments respectively, Supplementary Table 4). In other words, we compute for each species a composite matrix $P$ (i.e., a 1170 × 1170 matrix for the shallow coastal habitat and a 1163 × 1163 matrix for the neritic shelf habitat, encompassing both sampled and non-sampled localities) that fits best its dispersal traits, averaging ten years of realistic current-driven dispersal in the Mediterranean Sea. Single-generation dispersal estimates are directly extracted from one of these composite matrices. As explained next, multi-generational dispersal estimates are finally obtained by applying additional computations on these composite matrices.

## Cumulating implicit and explicit links in multi-generation dispersal models

Explicit links[44] evaluate filial connectivity from parents to children, assuming unique lineage (Fig. 2b). It is the usual proxy of connectivity used in other multi-generation models to assess gene flow between localities[34–38,41–43]. Implicit links[44] evaluate coalescent connectivity among siblings with common parents, considering multiple lineages (Fig. 2c). To estimate multi-generation dispersal probabilities between all locality-pairs, considering explicit or implicit links, we apply the theoretical formulations described in Ser-Giacomi et al.[44] on composite matrices. The main novelties of this approach are (i) the adequate consideration of putative intermediate connections or steps between any of the non-sampled localities of both habitats (Fig. 1b, c) and (ii) the fact that it allows cumulating connectivity pathways over each consecutive generation, i.e., from one generation to a fixed number of generations. For example, the connection between two localities over three generations of dispersal also accounts for connections over two and one generations. Analytical formulations for both cumulative explicit and implicit probabilities for any number of generations are established in Ser-Giacomi et al.[44] and are theoretically bounded to 1 for an infinite number of generations.

The general expression of explicit dispersal probabilities over $G = 2$ generations based on the composite matrix $P$ is:

$$P^{G=2} = P + P(L \circ P) \qquad (2)$$

The circle denotes the Hadamard product, and $L$ is the all-ones matrix minus the identity matrix. When applying Eq. (2) for two localities A and B (example illustrated in Fig. 2b), explicit link cumulates: (i) the sampled locality-pair explicit probability ($P_{AB}$) and (ii) the products of probabilities between sampled localities and their second generation intermediate locality ($P_{Ak_1}P_{k_1B}$, Fig. 2b), that is:

$$P_{AB}^{G=2} = P_{AB} + P_{Ak_1}P_{k_1B} \qquad (3)$$

The general expression of implicit dispersal probabilities over $G = 2$ generations based on the composite matrix $P$ is:

$$P^{G=2} = P^T P + P \left[ L \circ \left( P^T P \right) \right]^T P \qquad (4)$$

As before, the circle denotes the Hadamard product, and $L$ is the all-ones matrix minus the identity matrix. When applying Eq. (4) for two localities A and B (example illustrated in Fig. 2c) implicit link cumulates: (i) the product of probability between sampled localities and their common source (i.e., parent) localities $\left( P_{Ak_1}P_{Bk_1} \right)$; and (ii) the product of probability between sampled localities and their common source localities through two generations ($P_{AI_A}P_{I_Ak_2}P_{BI_B}P_{I_Bk_2}$, Fig. 2c), that is:

$$P_{AB}^{G=2} = P_{Ak_1}P_{Bk_1} + P_{AI_A}P_{I_Ak_2}P_{BI_B}P_{I_Bk_2} \qquad (5)$$

The Hadamard product vanished in Eq. (3) and equation (5) because there is no self-loop (i.e., self-recruitment) in none of our simplified exemplary localities. If self-recruitment exists, e.g., if $k_1 = A$ in Fig. 2c, siblings are found in both origin and destination localities implying that implicit links also encompass explicit links. Since $F_{st}$ are theoretically symmetrical (i.e., $F_{st_{AB}}$ equals $F_{st_{BA}}$), explicit dispersal probabilities have been transformed following $1 - (1 - P_{AB})^*(1 - P_{BA})$ to be symmetrical. Note that implicit dispersal probabilities between locality-pairs are already symmetrical by construction[44]. Both multi-generation explicit and implicit dispersal models are computed for 1, 5, 10, 20, 40, 60, 80, 100, 150, 200, 300, 400 and 500 generations using species-specific composite matrices.

## Reciprocal transformation of dispersal probabilities into modelled $F_{st}$

We perform a reciprocal transformation of dispersal probabilities to compare them against $F_{st}^{obs}/(1 - F_{st}^{obs})$ with linear models. As such, dispersal probabilities are linearized and transformed into modelled $F_{st}$ ($F_{st}^{mod}$) following:

$$F_{st}^{mod} = \frac{1}{\alpha^* P_{AB}^G + \frac{1}{\max(F_{st}^{obs}) - \min(F_{st}^{obs})}} + \min\left( F_{st}^{obs} \right) \qquad (6)$$

where $P_{AB}^G$ is the symmetrical dispersal probability of connection between localities A and B for a fixed number of generations G (derived either from single-generation explicit or multi-generation explicit or implicit dispersal models); α is a coefficient modulating the reciprocal transformation. The sensitivity of $F_{st}^{mod}$ to a wide range of α value has been thoroughly tested in order to retain the optimal value for each study (see Supplementary Fig. 4 and 5). The maximal and minimal $F_{st}^{obs}$ values averaged across the meta-analysis are included so that (i) when the dispersal probability is null, the reciprocal transformation returns a $F_{st}^{mod}$ equals to the mean maximal $F_{st}^{obs}$ (i.e., the average of all maximal $F_{st}^{obs}$ values of the 58 compiled studies, which is 0.1414) and (ii) when the probability of connection is 1, the $F_{st}^{mod}$ tends toward the mean minimal $F_{st}^{obs}$ (i.e., the average of all minimal $F_{st}^{obs}$ values of the 58 compiled species, which is 0.0127). Note that our goal here is not to scale each study's reciprocal transformation of pairwise probabilities into $F_{st}^{mod}$ by its own extrema $F_{st}^{obs}$, but rather to optimize robustness across the meta-analysis using the mean maximal and minimal $F_{st}^{obs}$ over 58 population genetics studies.

## Screening for the best models predicting observed genetic differentiation

We test for the predictions of $F_{st}^{obs}/(1 - F_{st}^{obs})$ by $F_{st}^{mod}$ derived from multi-generation dispersal models for 1 to 500 generations for all studies of the meta-analysis using maximum-likelihood population-effects (MLPE) linear mixed models. In addition of enabling the cross-comparison of several models for their predictive abilities, this statistical approach accounts for non-independence of pairwise comparisons, a distinguished feature of $F_{st}$ between localities[8,34,36,68]. We also search for correlations between $F_{st}^{mod}$ and $F_{st}^{obs}/(1 - F_{st}^{obs})$ using classical Mantel tests (see Supplementary Tables 7, 8 and 9). For each study, we parametrize our models by (i) selecting the optimal α value for which

the model AIC displays the lowest value (Supplementary Figs. 4 and 5, Supplementary Tables 7, 8 and 9) and (ii) selecting the optimal number of generations for which the AIC displays the lowest value (Supplementary Figs. 6 and 7, Supplementary Tables 7, 8 and 9). In addition, we perform Isolation-By-Distance (IBD) analyses of all compiled genetic structures with two proxies of distance between locality-pairs: the Euclidian distance (i.e., straight-line geographical distances) and the sea-least-cost distance, which corresponds to the length of the shortest path considering only maritime areas. Sea-least-cost distances are calculated thanks to the Marmap package[90] (version 1.0.4) on R (version 4.0.2). For the IBD analyses, we consider a two-dimensional dispersal model and thus test for predictions of $F_{st}^{obs}/(1 - F_{st}^{obs})$ by $\log_e(\text{distance})$[48,54].

To estimate the quality (predictive power) of each model relative to each of the other models, we also test for predictions of $F_{st}^{obs}/(1 - F_{st}^{obs})$ by $F_{st}^{mod}$ obtained with the dispersal models (one-generation explicit, multi-generation explicit, and multi-generation implicit). We fit the five predictors to $F_{st}^{obs}/(1 - F_{st}^{obs})$ with MLPE linear mixed models, considering a random effect on the locality level, thanks to the 'lmer' function of the lmerTest package (version 3.1.3). The coefficient of determination $R^2$ is computed with the function 'r.squaredGLMM' of the MuMIn package (version 1.43.17). All these tests are computed using the R software (version 4.0.5). For each study, we compare MLPE model predictions to determine the best model ability to predict gene flow. More specifically, we compute for each study the relative likelihood $\exp((\text{AIC}_{min} - \text{AIC}_i)/2)$ to determine the probability of each model $i$ (IBDs, one-generation explicit, multi-generation explicit and multi-generation implicit) to minimize the information loss of being the best one. We cross-compare model predictions across the entire meta-analysis by computing (i) the relative likelihood for each model against the four others (i.e., $\text{AIC}_{min}$ being the minimal value among IBDs, one-generation explicit, multi-generation explicit, and multi-generation implicit models) and (ii) the non-symmetrical pairwise relative likelihood difference (i.e., $\text{AIC}_{min}$ being the minimal value among both models under comparison). For example, for models A and B whose relative likelihood are 1 and 0.7 respectively, the relative likelihood difference between A and B is 0.3 and the relative likelihood difference between B and A is −0.3.

We test the sensitivity of MLPE model results ($R^2$ and $p$-values) against species-specific (taxonomic group, PLD, spawning season) and studies-specific (marker, number of sampled localities, $D_{btw}$ and $F_{st}^{obs}$ range) factors with ANOVA and linear regressions (Supplementary Table 10). Sensitivity tests are performed using the Matlab software (version 9.4).

Throughout the entire manuscript, asterisks inform about the significance of all statistical tests, as follows: *≤ 0.05; **≤ 0.005; ***≤ 0.0005 and "ns" stands for statistically non-significant.

### Reporting summary
Further information on research design is available in the Nature Research Reporting Summary linked to this article.

## Data availability
The population genetic data generated in this study is provided as a Supplementary Information (Supplementary Data 1). Genetic population studies references are provided as a Supplementary Information, as well as literature references used to categorize species characteristics (Supplementary Methods 1). We also used FishBase (https://www.fishbase.se/search.php) and Doris (https://doris.ffessm.fr/) websites for global information about the species of interest. Source and raw data relevant for each figure are provided with this paper. Source data are provided with this paper.

## Code availability
The Python codes used to compute multi-generation explicit and implicit dispersal probabilities have been already published (Ser-Giacomi et al.[44]; https://doi.org/10.1103/PhysRevE.103.042309) and are available online here: https://github.com/serjaaa/cumulated-net-conn.

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

## Acknowledgements

T.L. is funded by a Doctoral fellowship obtained through Aix-Marseille University. T.L., A.C and V.R acknowledge financial support from the European project SEAMoBB (Solutions for sEmi-Automated Monitoring of Benthic Biodiversity), funded by ERA-Net Mar-TERA (id. 145) and managed by the ANR (Grant No. ANR-17-MART0001-02, P.I.: A.C.). V.R., E.S.-G., N.B. and S.A.-H. acknowledge financial support obtained through the HYDROGENCONNECT project (P.I. V.R.) funded by the French programme MISTRALS ENVI-Med. E.S.-G. thanks C. Payrató Borrás and S. Meloni for stimulating discussions on Network Theory. E.S.-G. is very grateful for support from the Simons Foundation: the Simons Collaboration on Computational BIOgeochemical modeling of Marine EcosystemS (CBIOMES #549931). T.L. and V.R. warmly thank Madiop Lo for the technical improvements implemented in the LFN model. T.L. thanks E. Boulanger for the advice on the mixed model implementation. V.R., A.C., S.A.-H. and N.B. thank the GDR iMarCo for the initiation of this work, with a particular mention to Barbara Porro and Neil Alloncle for their implications in the first steps of the data compilation.

## Author contributions

T.L. and V.R. planned and designed research with important contributions from A.C. and building upon earlier discussions with N.B. and S.A-H. T.L., A.C. and S.A-H. contributed substantially to the compilation of genetic population studies. E.S-G. developed the theoretical formulations of the multi-generation dispersal models. T.L. performed data analysis with important contributions from V.R. and A.C. The manuscript was written by T.L. and V.R., and A.C., E.S-G, S.A-H and N.B. provided important contributions and critical revisions. All authors approved the final version of this manuscript.

## Competing interests

The authors declare no competing interests.
