## [Peer Review File · Nature Communications]

Spatial coalescent connectivity through multi-generation dispersal modelling predicts gene flow across marine phylaREVIEWER COMMENTS

Reviewer #1 (Remarks to the Author):

Predicting gene flow patterns from simulations as done in the present work offer promising approaches to unveil eco-evolutionary forces shaping population differentiation. In species where connectivity is mainly driven by dispersal larval phases their modelling has been done considering only single generations.

The authors compare several models of dispersal (Euclidian distance, Sea least-cost distance, single-generation explicit, multi-generation explicit and multi-generation implicit) with the observed Fst distances reported in 58 population genetic studies. They found that multi-generation coalescent connectivity is significantly better and explain 50% of observed genetic differentiation variance.

The obtained results seem reasonable and evolutionary meaningful and merit their publication.

I provide some information that could be further clarified or discussed.

Why for dispersal models testing IBD, Euclidian and sea-least cost distances were chosen instead of the shortest distance following the coast line? How meaningful in biological terms for the species would be the comparison with these three models? This information could be provided and discussed

It seems to me that 100 propagules were simulated in each release and for each experiment repeated approximately 10 times per year (10-day periodicity) along 10 years. The number of Lagrangian experiments in Table SI-4 is related to that? Please rewrite for clarity.

Was the simulation carried out for the 1170 populations in the shallow coastal habitat and 1163 for the neritic shelf so that each connectivity matrix is considering those populations, or was it among the 8196 nodes? The composite matrix P for a given species in single generation dispersal estimates would contain the populations sampled in the genetic study? Please specify for clarity.

For estimating multi-generation dispersal probabilities considering explicit or implicit links which and how many are the putative intermediate non-sampled populations? One for each generation?

Although the modelling is described in a previous paper it would be interesting to explain it in more detailed in the methods section to improve the comprehension since it is key in the analysis.

Is the optimal M in table SI-1 the optimal number of generations? Explain what the headers are in all tables for clarity.

In the Mantel tests methods section it is indicated that 40 generations maximizes the significant Mantel correlations according to SI-1 and SI-4. This information is difficult to interpret from these two tables. Is it only referring to SI-1?

It is indicated that the optimal number of generations to best predict gene flow significantly correlates with the sampling coverage scaled by the species-specific dispersal abilities. Please clarify.

Why in Fig 5a for 15 populations the probability of significant gene flow prediction is 0? The last sentence in page 10, does this mean that the model implemented can only provide accurate results for widely distributed sampling designs?

Reviewer #2 (Remarks to the Author):

The study uses existing data on 47 marine organisms over the Mediterranean basin to test multi-generation and coalescent multi-generation dispersal models against more classic isolation-by-distance (IBD) and single-generation dispersal models. The results show that the multi-generation and coalescent multi-generation models explain a greater proportion of genetic variation. Furthermore, the multi-generation models provide the opportunity to explore the number of generations that maximise the fit between predicted and observed values of genetic differentiation. The large dataset

considered also allows to do this with respect to the spatial scale and number of populations considered. The results show that the numbers of generations relevant to link demographic and genetic connectivity are in the order of tens of generations, that a few tens of populations need to be sampled, and that these numbers depend on the pelagic larval duration of the species considered.

The manuscript is clear and well written. Assuming that the model published in reference 42 (Ser-Giacomi et al. 2021, which I did not review) is correct, the study appears to be sound. It is an important contribution because it contributes to improve our understanding of the link between demographic and genetic connectivity, which represents an important knowledge gap.

One aspect that I find paradoxical is that in the island model that is used to make the link between modeled dispersal and modeled genetic structure, dispersal is not spatially explicit (i.e. it is equally likely between any pair of population). So a classic model in which dispersal is explicit but not spatially explicit is used to test a spatially explicit model that forcefully shows that space is important when interpreting genetic structure. How can this conundrum be resolved? Also the island model assumes well-defined populations but the "populations" considered in the model (black and green squares in Figure 1b and c) are clearly not discrete populations.

The study shows that the multi-generation and coalescent and multi-generation dispersal models outperform the IBD and single-generation models but Figure 3 suggests that the coalescent aspect only represents a slight improvement to the multi-generation model. In which cases/situations is the coalescent aspect most important?

The model assumes symmetric dispersal but dispersal is probably highly asymmetric (the oceanographic model can be used to address this in detail). How is that expected to affect the results?

Another point is that this approach requires a sophisticated distributional and oceanographic model of the study area. In the absence of such a model this approach cannot be implemented. This is important to remind.

Finally, considering not the general approach but the specific case of the Mediterranean Sea: did the study reveal any new pattern or process?

Minor comments

Statement of authorship: state NB's contribution

Abstract

Replace "other multiple generations" by "over multiple generations"

Introduction

"While a small proportion of migrants could be sufficient to ensure gene flow between distant populations": precise that this is considering an infinite island model.

Results

"for phylogenetically divergent 47 marine species ": rephrase

Discussion

replace "so that it can be readily apply" by "so that it can be readily applied"

"About one third of the compiled studies displayed significant IBD predictions with a mean Mantel R^2 " clarify what is meant by "with a mean Mantel R^2 "

Results

"i.e filial connectivity, Fig 2b": add missing period.

Reviewer #3 (Remarks to the Author):

Summary

The authors present a meta-analysis of 58 population genetic studies from the Mediterranean basin in which they use new distance measures based on connectivity probability graphs from a Lagrangian biophysical model to explain F_{st} among populations. The distances they use are the cumulative products of multigenerational dispersal through the graphs; both an explicit distance based on dispersal from parents and an implicit distance which includes dispersal by siblings are calculated. Both of these multigenerational distances from a biophysical model have a higher mean Mantel R^2 with observed F_{st} than more traditional distances such as Euclidean or overwater distance, with the implicit distance having the highest mean correlation. Interestingly, genetic sampling strategy is found to be predictive of a significant correlation.

Major Comments

This is an interesting study, and I'm quite excited that the authors have taken graph theory the extra generational steps beyond what others have done to derive these distances and show that they generally do a better job in explaining observed F_{st} across a decently large sample of species. However I have a number of reservations about their methods.

1) The authors use Wright's classic formula based on an island model (equal population sizes and equal migration rates among all demes) to convert migration probabilities from their biophysical model into F_{st} . This simply doesn't make sense. First, we are clearly not in an island model and both the biophysical model and the empirical data clearly violate most of its assumptions (Whitlock and McCauley 1998). Second, the m in this conversion is the proportion of migrants, or more specifically to the system, the proportion of individuals that send a migrant to another deme. This is not analogous to the dispersal probability estimated from the biophysical model and corrected for multiple generations etc. Finally, the use of a constant and low (for marine populations) N_e just amplifies the violation of the model's assumptions of equal N_e . Fluctuation in local N_e is probably the largest component in the mismatch between F_{st} and various models (Faurby and Barber 2012). A possible alternative is to simply look for correlations between observed F_{st} and the different corrections to dispersal probability.

2) Lagrangian methods for biophysical modeling get a lot of attention in seascape genetics because they more realistically model particle movement, but where they fall down is in the relatively small number of particles that they can model. This is important in population genetic applications because F_{st} is sensitive to a small number of long-distance dispersal events. 100 particles for >1000 demes is fine for a Lagrangian model, but it comes nowhere close to modeling the actual number of larvae released and it probably misses nearly all of the long-distance dispersal events in the Mediterranean system. I'm not sure what can be done with the manuscript at this stage, but at least a healthy paragraph of discussion is warranted.

3) As described by Guillot & Rousset (2013), Mantel tests are not appropriate for comparing two matrices that are both autocorrelated because it will give a high rate of false positives. See Wagner and Fortin (2015) for possible alternatives.

Specific Comments

Title and throughout - While I understand that the word "coalescent" is used correctly here, it runs the risk of confusing readers (as it did me) that it is referring to the coalescent theory of population genetics. In this paper, it is the spatial model that is coalescent, not the genetic model.

L88 - "While a small proportion of migrants could be sufficient to ensure gene flow between distant populations 7, the inherent spatial scales of genetic structures are generally a few orders of magnitude higher than potential dispersal distances over a single generation 25, even for species exhibiting extremely rare long-distance dispersal 26,27."

Not sure I follow this sentence, which may just be a grammatical thing. Are you saying that in the ocean, the spatial scale at which populations are structured is orders of magnitude greater than dispersal distances? I would agree.

L110 - connections

L111 - Seldom used

L140 - "cumulating those ensured by previous..." - I suggest: accounting for those accumulated by previous....

L146 - "observed genetic structures" (and throughout the manuscript) This is grammatically correct, but usage in the population genetics community is to have "structure" as singular.

L174 - what is meant by least-cost in this context? What is the cost? Is this shortest overwater distance?

L333 - What specifically was extracted from the 58 studies? Was it pairwise Fst? Or were the actual data-reanalyzed? If the former, there are a lot of different estimators of Fst... how was this standardized among studies?

L343,356, 375, 377 etc. - the term "populations" is difficult to define in this context. See e.g. Waples and Gaggiotti (2006). "Locality" or "deme" would be better terms.

L376 - 100 propagules per population is quite low!!

L381 - It should be made clear that the 5 PLDs are the 5 bins that were created for the empirical studies.

L406 - cumulative

L262 and throughout - because this journal uses numerical citations, author names and year need to be cited when using them in a sentence.

L270 "which is comparable to a previous meta-analysis" - also comparable to Selkoe and Toonen 2011.

L276 "Single-generation dispersal models are worse than IBD models to predict genetic connectivity probably because IBD is supported by robust theory." - this seems like an oversimple explanation. The values from a biophysical model are just another distance after all.

L312 "Moreover, *island model theory* assumes..."

Figure 3 - I don't understand what is being depicted with the circles. The reference model minus the alternate model? I don't understand why... Also, how were confidence intervals determined? Finally, the text refers to Fisher's

combined probability being depicted here, but I can't find it.

Figure 4 - Color ramps seem backwards to me in panels b and c

Figure 5 - I don't think I understand how this figure was created. Why aren't there 58 points? Why is only one point at zero when in fact ~60% should be at zero? I don't think it is valid to bin by sample size and then fit a logistic model...

Literature Cited

Waples RS, Gaggiotti O. 2006. What is a population? An empirical evaluation of some genetic methods for identifying the number of gene pools and their degree of connectivity. *Mol Ecol.* 15(6):1419–1439. doi:10/fkpz2w.

Guillot G, Rousset F. 2013. Dismantling the Mantel tests. Harmon L, editor. *Methods Ecol Evol.* 4(4):336–344. doi:10.1111/2041-210x.12018.

Selkoe KA, Toonen RJ. 2011. Marine connectivity: a new look at pelagic larval duration and genetic metrics of dispersal. *Mar Ecol Prog Ser.* 436:291–305. doi:10.3354/meps09238.

Wagner HH, Fortin MJ. 2015. Basics of Spatial Data Analysis: Linking Landscape and Genetic Data for Landscape Genetic Studies. In: Balkenhol N, Cushman SA, Storfer AT, Waits LP, editors. Chichester, UK: John Wiley & Sons, Ltd.

Whitlock MC, McCauley DE. 1999. Indirect measures of gene flow and migration: F_{ST} not equal $1/(4Nm+1)$. *Heredity.* 82:117–125. doi:10/d22rz5.

REVIEWER COMMENTS

Reviewer #1 (Remarks to the Author):

Predicting gene flow patterns from simulations as done in the present work offer promising approaches to unveil eco-evolutionary forces shaping population differentiation. In species where connectivity is mainly driven by dispersal larval phases their modelling has been done considering only single generations.

The authors compare several models of dispersal (Euclidian distance, Sea least-cost distance, single-generation explicit, multi-generation explicit and multi-generation implicit) with the observed F_{st} distances reported in 58 population genetic studies. They found that multi-generation coalescent connectivity is significantly better and explain 50% of observed genetic differentiation variance. The obtained results seem reasonable and evolutionary meaningful and merit their publication.

Thank you for the positive appreciation and for your interests in our work.

Why for dispersal models testing IBD, Euclidian and sea-least cost distances were chosen instead of the shortest distance following the coast line? How meaningful in biological terms for the species would be the comparison with these three models? This information could be provided and discussed

Our paper does not intend to test IBD by itself; conversely, previous models of spatial genetic structures short-listed here are cross-compared to show that our novel models perform better than classical approaches. Here we retain the well-documented “sea-least cost distances” instead of the “shortest distance following the coastline” to avoid redundancy (in the absence of islands, both measures would be very similar) and since the latter is less accurate than the former. This said, we acknowledge (in *Discussion*, l.321-326) that these methods have little biological meaning as they only relate to geographical distances without considering properly the complex movements of individuals. Through the development and application of our multi-generation dispersal models simulating plainly propagules' movements across the seascape, we hope to limit the shortcomings of IBD models in the near future.

It seems to me that 100 propagules were simulated in each release and for each experiment repeated approximately 10 times per year (10-day periodicity) along 10 years. The number of Lagrangian experiments in Table SI-4 is related to that? Please rewrite for clarity.

This is correct, we implemented 100 propagules in each population, now referred to as localities (i.e. the green squares displayed in Figure 1b,c). There are approximately 1,200 localities in each habitat, meaning we tracked approximately 120,000 propagules per Lagrangian experiment. The Table SI-4 refers to the number of Lagrangian experiments aggregated for each spawning season. Finally, dispersal probabilities associated to each study were obtained by integrating about 12,000,000 (seasonal spawning) to 48,000,000 (annual spawning) propagules trajectories. The main text (in *Methods*, subsection “Biophysical Modelling”) and the caption of Table SI-4 were rewritten to improve clarity.

Was the simulation carried out for the 1170 populations in the shallow coastal habitat and 1163 for the neritic shelf so that each connectivity matrix is considering those populations, or was it among the 8196 nodes? The composite matrix P for a given species in single generation dispersal estimates

would contain the populations sampled in the genetic study? Please specify for clarity. For estimating multi-generation dispersal probabilities considering explicit or implicit links which and how many are the putative intermediate non-sampled populations? One for each generation?

We considered all the populations, now referred to as localities, delineated by both habitats (i.e. 1170 and 1163 localities for the shallow coastal and neritic shelf habitats, respectively) in the estimation of multi-generation dispersal probabilities. The composite matrix P (e.g. equivalent to single generation dispersal) for a given species does contain the dozens of localities sampled in the studies, along with all the remaining non-sampled ones. In other words, there are ~ 1200 putative intermediate non-sampled localities when estimating filial or coalescent connectivity between two sampled localities. It has been clarified in the revised manuscript (l. 441 to l. 451).

Although the modelling is described in a previous paper it would be interesting to explain it in more detailed in the methods section to improve the comprehension since it is key in the analysis.

While the method is fully described in Ser-Giacomi et al., (2021), we made additional efforts to explain it here in a clear, yet simple, manner without duplicating already published work. In particular, Fig. 2 presents well-thought schematics to visualize how explicit and implicit links have been computed. In addition, we rewrote the subsection “Cumulating implicit and explicit links in multi-generation dispersal models” in *Methods* (including simplified equations with intelligible examples) to improve clarity (l. 452 to l.496).

Is the optimal M in table SI-1 the optimal number of generations? Explain what the headers are in all tables for clarity.

Yes, it is correct. We added details (now in Tables SI-7, SI-8, SI-9 of the revised SI) to clarify this doubt. Note that we now employ MLPE linear mixed models to test for the predictions of *observed Fst* by different approaches (see responses to reviewers #2 and #3); consequently, the AIC (Akaike Information Criterion) allows assessing the quality of each model (each generation, in this case) to select the optimal generation.

In the Mantel tests methods section it is indicated that 40 generations maximizes the significant Mantel correlations according to SI-1 and SI-4. This information is difficult to interpret from these two tables. Is it only referring to SI-1?

We rewrote this section since we now used MLPE linear mixed models rather than Mantel tests. We also added Fig. SI-7 to improve comprehension.

It is indicated that the optimal number of generations to best predict gene flow significantly correlates with the sampling coverage scaled by the species-specific dispersal abilities. Please clarify.

We rewrote this section since we now used MLPE linear mixed models rather than Mantel tests.

Why in Fig 5a for 15 populations the probability of significant gene flow prediction is 0?

We binned the studies considering the number of populations sampled, so each point in this figure corresponds to a particular number of sampled populations (in the words of population genetic studies). Across the meta-analysis, only one study (i.e. Marzouck et al., 2017) used 15 populations. In

this study, two genetic markers were used to assess F_{st} between sampled populations: a nuclear marker and a mtDNA marker, both characterized by nucleotide sequences. This study focuses on a mollusk, *Hexaplex trunculus*, which has the particularity to aggregate to spawn large egg masses (~ 10 cm) that then undergo intracapsular development (no larval phase indeed). This could alter its dispersal by currents and may thus explain why our model returns inconsistent gene flow predictions for both markers. Note that this figure is no longer in the revised main manuscript.

The last sentence in page 10, does this mean that the model implemented can only provide accurate results for widely distributed sampling designs?

We agree with the reviewer on this point: in this study and in all the population genetics analysis, the ability of our multi-generation dispersal model to accurately predict gene flow is influenced by the sampling strategy. When validating our methodology through correlations with *observed* F_{st} , it clearly shows that widely distributed sampling design gives more power (i) to evaluate effectively genetic differentiation over the species broad-scale distribution and (ii) to predict accurately gene flow (i.e. more points in the mixed model). Low correlations can thus arise either from the model itself or from sampling deficiencies, and there is no statistical way to objectively tease those two hypotheses apart.

Reviewer #2 (Remarks to the Author):

The study uses existing data on 47 marine organisms over the Mediterranean basin to test multi-generation and coalescent multi-generation dispersal models against more classic isolation-by-distance (IBD) and single-generation dispersal models. The results show that the multi-generation and coalescent multi-generation models explain a greater proportion of genetic variation. Furthermore, the multi-generation models provide the opportunity to explore the number of generations that maximise the fit between predicted and observed values of genetic differentiation. The large dataset considered also allows to do this with respect to the spatial scale and number of populations considered. The results show that the numbers of generations relevant to link demographic and genetic connectivity are in the order of tens of generations, that a few tens of populations need to be sampled, and that these numbers depend on the pelagic larval duration of the species considered.

The manuscript is clear and well written. Assuming that the model published in reference 42 (Ser-Giacomi et al. 2021, which I did not review) is correct, the study appears to be sound. It is an important contribution because it contributes to improve our understanding of the link between demographic and genetic connectivity, which represents an important knowledge gap.

Thank you for the positive comments; we are glad that you consider this work promising and impacting as it bridges an important knowledge gap. We also hope that, once published, it will help the research community to better comprehend the causes and implications of both demographic and genetic connectivity.

One aspect that I find paradoxical is that in the island model that is used to make the link between modeled dispersal and modeled genetic structure, dispersal is not spatially explicit (i.e. it is equally likely between any pair of population). So a classic model in which dispersal is explicit but not spatially explicit is used to test a spatially explicit model that forcefully shows that space is important when interpreting genetic structure. How can this conundrum be resolved?

We had initially used the island model to transform our multi-generation dispersal probabilities into genetic distances and then used Mantel test with Pearson correlation method to test for correlations with observed genetic differentiation. Despite the caveats intrinsically linked to the island model theory (as documented by the reviewer), we had chosen this model because it corresponds to a reciprocal transformation of dispersal probabilities into genetic distances. As the relationship between F_{st} and $N_e m$ was merely interpreted, any reciprocal conversion would work fine. Following this relevant comment, we now use a reciprocal transformation of dispersal probabilities; we rewrote accordingly the sections *Methods*, *Results* and SI-V in the revised manuscript. There is indeed no consensus on how to transform “dispersal probabilities” or “oceanographic distances” into “genetic distances”. Some studies used a \log_{10} transformation (e.g. Crandall et al., 2012; Jahnke et al., 2018), which however does not allow handling null probabilities. We also tested (not shown) Reynold’s distance $\log_{10}(1-F_{st})$, which (i) returned worse results than those obtained with the reciprocal transformations and (ii) does not allow comparing models with AIC. Last but not least, we use in the revised manuscript $F_{st}/(1-F_{st})$ instead of F_{st} when testing for predictions of our five dispersal models to emphasize that fact the island model has been replaced by a more accurate framework (Rousset, 2001).

Also the island model assumes well-defined populations but the "populations" considered in the model (black and green squares in Figure 1b and c) are clearly not discrete populations.

We do not rely anymore on the island model in the revised manuscript so that this comment is now outdated. Nevertheless, the reviewer is right for the physical assumptions of our models: it evaluates multistep connectivity across an ensemble of contiguous, yet discrete, sampled, or non-sampled populations of a given habitat (now called localities). It may thus appear quite different from the discrete and almost-isolated populations assumed by the island model. We made this choice because broad-scale spatialized data of both substrate and species distribution do not exist or are not precise and homogeneous enough over the Mediterranean Sea to properly delineate discrete populations for each species of the meta-analysis. It is perhaps one of the main difficulties of investigating connectivity over temperate seascapes (characterized by relatively long and continuous coastlines with diverse and patchy habitats) as compared to tropical seascapes (characterized by small isolated islands and less variable habitats) such as in the South Pacific (Crandall et al., 2012) or the Caribbean Sea (Kool et al., 2010).

The study shows that the multi-generation and coalescent and multi-generation dispersal models outperform the IBD and single-generation models but Figure 3 suggests that the coalescent aspect only represents a slight improvement to the multi-generation model. In which cases/situations is the coalescent aspect most important?

The coalescent model is conceptually the most natural way of evaluating gene flow between two contemporary sampled populations (i.e. the two sampled populations share the same temporality, contrary to filial connectivity which considers both populations isolated by the number of generations considered, see Figure 2b,c). Moreover, in the case of self-recruitment, filial connectivity is a particular case of coalescent connectivity (e.g. when $A = k_2$ in Figure 2c). In other words, and considering that simulated self-recruitment is prominent, filial connectivity appears included within coalescent connectivity. Then, (i) when explicit links are dominant, implicit links would not add much information (i.e. coalescent connectivity is equivalent to filial connectivity), (ii) when explicit links are weak, implicit links can be strong. The latter situations are those where coalescent connectivity largely improves any evaluation solely based on filial connectivity.

The model assumes symmetric dispersal but dispersal is probably highly asymmetric (the oceanographic model can be used to address this in detail). How is that expected to affect the results?

The reviewer is right, dispersal is, in nature, highly asymmetric. However, F_{st} are symmetric. Hence, we had to transform asymmetric dispersal probabilities between two populations (e.g. P_{AB} from A to B and P_{BA} from B to A in Fig. 2b) into a unique and symmetric metric evaluating connection probabilities thanks to current-driven dispersal. This transformation would not be necessary if we compiled in our meta-analysis directional migration values obtained by other genetic analysis methods (e.g. DivMigrate or GENECLASS2, Jahnke et al., 2018). However, these methods are not as ubiquitous as F_{st} , preventing the global synthesis achieved through our meta-analysis.

Note that this asymmetric to symmetric transformation applies to filial connectivity only; indeed, coalescent connectivity is symmetric by construction (Ser-Giacomi et al., 2021). The strength of this transformation is to remain in a probabilistic framework: we do not compute a mean or select the max/min value between P_{AB} - P_{BA} , but we rather look for the maximized probability of connection between two populations for a given number of generations. For example, if $P_{AB} = 0.9$ and $P_{BA} = 0.1$; the symmetric probability will be 0.91. Or if $P_{AB} = 0.5$ and $P_{BA} = 0.5$; the symmetric probability will be 0.75.

Another point is that this approach requires a sophisticated distributional and oceanographic model of the study area. In the absence of such a model this approach cannot be implemented. This is important to remind.

The reviewer is totally right, any estimation of currents-mediated connectivity (i.e. single generation dispersal, multi-generation dispersal using either explicit or implicit connections) requires velocity fields from oceanographic models. Nowadays, operational ocean models have been developed for many oceanic systems at both regional and global scales and many products are available for the scientific communities. As an example, we used velocity fields for the Mediterranean Sea available on <https://resources.marine.copernicus.eu/products>, but other oceanic domains are available (i.e. Baltic sea, Antarctic ocean, Atlantic ocean, up to the global ocean).

Finally, considering not the general approach but the specific case of the Mediterranean Sea: did the study reveal any new pattern or process?

In accord with the scope of the target journal and its international readership, we focus the writing on the general results that are understandable by anyone and are applicable anywhere, without going into details on the Mediterranean Sea. In other words, the Mediterranean Sea is here used a natural laboratory to investigate how to best simulate gene flow. Note that in Ser-Giacomi et al., (2021), we reported that transport barriers (often invoked in Mediterranean population genetic studies) are indeed permeable to implicit connections, notably the Oran-Almeria front and the Balearic front, suggesting that transport barriers are permeable to gene flow. From a genetic perspective, our meta-analysis showed that the southeastern Mediterranean shorelines are largely under-sampled, preventing a global comprehension of basin-scale genetic differentiation patterns. As said in the manuscript, the coalescent connectivity concept offers great perspectives to further investigate evolutive processes over the Mediterranean Sea.

Minor comments

Statement of authorship: state NB's contribution

Thanks, it has been modified in the revised manuscript.

Abstract

Replace "other multiple generations" by "over multiple generations"

Thanks, it has been modified in the revised manuscript.

Introduction

"While a small proportion of migrants could be sufficient to ensure gene flow between distant populations": precise that this is considering an infinite island model.

Thanks, the corresponding sentence has been rewritten in the revised manuscript.

Results

"for phylogenetically divergent 47 marine species ": rephrase

Thanks, it has been modified in the revised manuscript.

Discussion

replace "so that it can be readily apply" by "so that it can be readily applied"

Thanks, it has been modified in the revised manuscript.

"About one third of the compiled studies displayed significant IBD predictions with a mean Mantel R²" clarify what is meant by "with a mean Mantel R²"

Thanks, it has been detailed in the revised manuscript.

Results "i.e filial connectivity, Fig 2b": add missing period.

Thanks, it has been modified in the revised manuscript.

Reviewer #3 (Remarks to the Author):

The authors present a meta-analysis of 58 population genetic studies from the Mediterranean basin in which they use new distance measures based on connectivity probability graphs from a Lagrangian biophysical model to explain F_{st} among populations. The distances they use are the cumulative products of multigenerational dispersal through the graphs; both an explicit distance based on dispersal from parents and an implicit distance which includes dispersal by siblings are calculated. Both of these multigenerational distances from a biophysical model have a higher mean Mantel R^2 with observed F_{st} than more traditional distances such as Euclidean or overwater distance, with the implicit distance having the highest mean correlation. Interestingly, genetic sampling strategy is found to be predictive of a significant correlation.

This is an interesting study, and I'm quite excited that the authors have taken graph theory the extra generational steps beyond what others have done to derive these distances and show that they generally do a better job in explaining observed F_{st} across a decently large sample of species. However I have a number of reservations about their methods.

Thank you very much for your interests in our work and for the constructive comments. We have carefully addressed all your reservations below, which helped us to further improve the quality of our manuscript.

1) The authors use Wright's classic formula based on an island model (equal population sizes and equal migration rates among all demes) to convert migration probabilities from their biophysical model into F_{st} . This simply doesn't make sense. First, we are clearly not in an island model and both the biophysical model and the empirical data clearly violate most of its assumptions (Whitlock and McCauley 1998). Second, the m in this conversion is the proportion of migrants, or more specifically to the system, the proportion of individuals that send a migrant to another deme. This is not analogous to the dispersal probability estimated from the biophysical model and corrected for multiple generations etc. Finally, the use of a constant and low (for marine populations) N_e just amplifies the violation of the model's assumptions of equal N_e . Fluctuation in local N_e is probably the largest component in the mismatch between F_{st} and various models (Faurby and Barber 2012). A possible alternative is to simply look for correlations between observed F_{st} and the different corrections to dispersal probability.

That is a good point indeed (also raised by reviewer #2) that we have taken into account. As reported above, we previously used the inverse function of the island model to transform our multi-generation dispersal probabilities into genetic distances, without really exploiting the relationship between F_{st} and $N_e m$. Following these valuable comments, the island model was disregarded from the revised manuscript, which now uses different models of reciprocal transformations (see *Methods, Results* and section SI-5 in the revised manuscripts).

2) Lagrangian methods for biophysical modeling get a lot of attention in seascape genetics because they more realistically model particle movement, but where they fall down is in the relatively small number of particles that they can model. This is important in population genetic applications because F_{st} is sensitive to a small number of long-distance dispersal events. 100 particles for >1000 demes is fine for a Lagrangian model, but it comes nowhere close to modeling the actual number of larvae released and it probably misses nearly all of the long-distance dispersal events in the Mediterranean

system. I'm not sure what can be done with the manuscript at this stage, but at least a healthy paragraph of discussion is warranted.

The reviewer raises an interesting point that motivated us to do further analyses to test the sensitivity of long-distance dispersal events to the number of particles released in our biophysical model. To do so, we compared synthetic dispersal kernels over a single generation for all the sampled populations of the shallow coastal habitat using 100 (as in the manuscript) and 1000 particles per node. Dispersal kernels relate the distance of all connected populations to any sampled population with the probability of the connected populations to act as the sources of the sampled populations (as shown in Figure R1a,b). The binned comparison between both dispersal kernels (i.e. the probabilities associated to each distance categories, Fig. R1c) showed that long dispersal events are not sensitive to the number of particles modelised (Fig. R1d; $r = 0.9998^{***}$).

We recognize that rare long-distance dispersal events could shape the patterns of genetic differentiation through density-dependent processes (e.g. gene surfing, see Waters et al., 2013), and is more likely related to exceptionally long PLDs (rare event for which biological knowledge is sparse) than numerical restrictions. It could be very interesting to investigate the impact of these rare long-dispersal events on genetic differentiation in further work.

*Figure R1: Sensitivity of long-distance dispersal events to the number of simulated particles. Test is done for one exemplary dispersal event (spawning date on 01/06/2012) for a 30 days PLD case study species. **a,b** Dispersal plume and associated connection probabilities from the most southern sampled population (red contour) using **a** 100 particles and **b** 1000 particles. **c** Synthetic dispersal kernels for the 559 sampled populations using 100 (blue curve) and 1000 particles (red curve). **d** Comparison between the dispersal kernel computed with 100 particles and the dispersal kernel with 1000 particles ($r = 0.9998^{***}$). Distances from source nodes to sampled nodes were binned into 50 km classes from 0 to 800 km.*

3) As described by Guillot & Rousset (2013), Mantel tests are not appropriate for comparing two matrices that are both autocorrelated because it will give a high rate of false positives. See Wagner and Fortin (2015) for possible alternatives.

The reviewer raises a relevant issue that we carefully considered. Indeed, the usefulness of Mantel tests in sea/landscape genetics has been questioned in the last decade (e.g. Guillot & Rousset, 2013; Selkoe et al., 2016). Note however that the caveat of Mantel tests (e.g. inflating type I error rate) impacts primarily “artificial” dissimilarity matrices computed from spatialized data, such as environmental distances between two populations computed from gridded environmental datasets (e.g. Isolation-By-Environment, Wang et al., 2013), but not “pure” dissimilarity matrices that can be directly formulated in terms of distances (i.e. genetic distance or *Fst*, geographic distance or oceanographic/dispersal distance, Legendre & Fortin 2010, Legendre et al., 2015). This is why Guillot & Rousset (2013) stated: “The simple Mantel test is therefore suitable to test the absence of IBD from population genetic data in this case”. Thus, previously used Mantel tests were appropriate for our analysis because all matrices are “pure” dissimilarity matrices.

Nevertheless, and as suggested by the reviewer, seascape genetics must move beyond Mantel tests (Selkoe et al., 2016). We took this opportunity to repeat all our analyses using mixed models, such as maximum-likelihood population effects (MLPE) models, as a robust alternative to Mantel test since it permits to evaluate predictors of pairwise population genetic differentiation while accounting for non-independence of pairwise comparisons (e.g. Selkoe et al., 2016, Boulanger et al., 2019, Jahnke & Jonsson, 2022). Moreover, using mixed models allow comparing gene flow predictors (IBDs and single- or multi-dispersal models using either explicit or implicit connections) using *relative likelihood* computed with AIC. The latter properties is very useful in our work so that it has been retained, following this constructive reviewer’s suggestions, returning results overall clearer than before (e.g. ~70% instead of 50% of the variance explained).

Specific Comments

Title and throughout - While I understand that the word "coalescent" is used correctly here, it runs the risk of confusing readers (as it did me) that it is referring to the coalescent theory of population genetics. In this paper, it is the spatial model that is coalescent, not the genetic model.

Even if it could add minor confusion, we think the use of the term “coalescent” is judicious to define our novel dispersal model with implicit connections while linking biophysical models with genetic theory. It also lays the emphasis on this concept mostly disregarded by the marine connectivity community (that remains to-date focused on filial connectivity). Thanks to this relevant suggestion, we now clearly stated in the revised manuscript (l.137) that we refer to the dispersal model, not the genetic one.

L88 - "While a small proportion of migrants could be sufficient to ensure gene flow between distant populations 7, the inherent spatial scales of genetic structures are generally a few orders of magnitude higher than potential dispersal distances over a single generation 25, even for species exhibiting extremely rare long-distance dispersal 26,27."

Not sure I follow this sentence, which may just be a grammatical thing. Are you saying that in the ocean, the spatial scale at which populations are structured is orders of magnitude greater than dispersal distances? I would agree.

Yes, this is exactly what we wanted to address with this sentence (that has been now slightly rewritten).

L110 – connections

Thanks, it has been modified in the revised manuscript.

L111 - Seldom used

Thanks, it has been modified in the revised manuscript.

L140 - "cumulating those ensured by previous..." - I suggest: accounting for those accumulated by previous....

Thanks, it has been modified in the revised manuscript.

L146 - "observed genetic structures" (and throughout the manuscript) This is grammatically correct, but usage in the population genetics community is to have "structure" as singular.

Thanks, it has been modified in the revised manuscript.

L174 - what is meant by least-cost in this context? What is the cost? Is this shortest overwater distance?

Yes, least-cost refers to the shortest overwater distance, we used the same semantic as McRae & Beier, 2007. We added a precision in the revised manuscript.

L333 - What specifically was extracted from the 58 studies? Was it pairwise F_{st} ? Or were the actual data-reanalyzed? If the former, there are a lot of different estimators of F_{st} ... how was this standardized among studies?

Since it is practically impossible to obtain all raw data from each historical study, re-computing all F_{st} values was not feasible. Pairwise F_{st} estimates were extracted from the 58 studies except for a few cases (one or two) where we obtained raw data and computed them (or separated outlier and non-outlier SNPs before computing F_{st}). However, the design of our approach is robust and does not require standardization because there are no comparisons of F_{st} values among studies. Indeed, each observational study is treated individually when compared to biophysical models, prior to any global analysis of the congruence between genetic data and biophysical models.

L343,356, 375, 377 etc. - the term "populations" is difficult to define in this context. See e.g. Waples and Gaggiotti (2006). "Locality" or "deme" would be better terms.

Thanks, we replaced "population" by "locality" in the revised manuscript.

L376 - 100 propagules per population is quite low!!

See above our answer to a similar comment.

L381 - It should be made clear that the 5 PLDs are the 5 bins that were created for the empirical studies.

Thanks, it has been detailed in the revised manuscript.

L406 – cumulative

It has been rewritten.

L262 and throughout - because this journal uses numerical citations, author names and year need to be cited when using them in a sentence.

Thanks, it has been modified in the revised manuscript.

L270 "which is comparable to a previous meta-analysis" - also comparable to Selkoe and Toonen 2011.

Thanks, it has been added in the revised manuscript.

L276 "Single-generation dispersal models are worse than IBD models to predict genetic connectivity probably because IBD is supported by robust theory." - this seems like an oversimple explanation. The values from a biophysical model are just another distance after all.

Thanks, it has been detailed in the revised manuscript.

L312 "Moreover, *island model theory* assumes..."

Thanks, it has been detailed in the revised manuscript.

Figure 3 - I don't understand what is being depicted with the circles. The reference model minus the alternate model? I don't understand why...

Also, how were confidence intervals determined? Finally, the text refers to Fisher's combined probability being depicted here, but I can't find it.

Circles or dots indicate the mean Δ Mantel R^2 (Mantel R^2 of the left model minus Mantel R^2 of the right model) to perform pair-wise comparisons among Pearson correlation coefficients obtained with Mantel tests. The 95 % confidence interval is determined by the multiplication of the standard error ($\text{std}(\Delta \text{ Mantel } R^2)/(\text{Nbr of studies} - 1)$) with the 2.5th and 97.5th percentile of the Student's t distribution with $(\text{Nbr of studies} - 1)$. The Fisher's combined probabilities are reported by asterisks when significant (*, ** or ***) or by "ns" otherwise.

In the revised manuscript, we improve and clarify this figure. Squares indicate mean MLPE linear mixed model R^2 (only for significative R^2). Diamonds indicate the *relative likelihood* of the left model vs all models at the meta-analysis scale (considering the 58 studies). Dots indicate the *relative likelihood difference* of the left model vs the right model at the meta-analysis scale (considering the 58 studies).

Figure 4 - Color ramps seem backwards to me in panels b and c

We intentionally use inverse color-scale to ease visual understanding and best represent that high value of *Fst* intuitively represents low connectivity between population pairs.

Figure 5 - I don't think I understand how this figure was created. Why aren't there 58 points? Why is only one point at zero when in fact ~60% should be at zero? I don't think it is valid to bin by sample size and then fit a logistic model...

In brief, we binned the 58 studies according to their number of populations sampled. For each “number of populations sampled categories”, we account for the number of significant studies among the number of non-significant studies to display a probability of significant gene flow prediction. We already addressed (see a previous answer to reviewer #1) the specific case of the null prediction for the 15 populations of Marzouck et al. (2017). Jenkins et al., 2010 used the same methodology for their Figure 2. Note that this figure is no longer in the revised main manuscript.

Literature Cited by Reviewer #3

Waples RS, Gaggiotti O. 2006. What is a population? An empirical evaluation of some genetic methods for identifying the number of gene pools and their degree of connectivity. *Mol Ecol.* 15(6):1419–1439. doi:10/fkpz2w.

Guillot G, Rousset F. 2013. Dismantling the Mantel tests. Harmon L, editor. *Methods Ecol Evol.* 4(4):336–344. doi:10.1111/2041-210x.12018.

Selkoe KA, Toonen RJ. 2011. Marine connectivity: a new look at pelagic larval duration and genetic metrics of dispersal. *Mar Ecol Prog Ser.* 436:291–305. doi:10.3354/meps09238.

Wagner HH, Fortin MJ. 2015. Basics of Spatial Data Analysis: Linking Landscape and Genetic Data for Landscape Genetic Studies. In: Balkenhol N, Cushman SA, Storfer AT, Waits LP, editors. Chichester, UK: John Wiley & Sons, Ltd.

Whitlock MC, McCauley DE. 1999. Indirect measures of gene flow and migration: F_{ST} not equal $1/(4Nm+1)$. *Heredity.* 82:117–125. doi:10/d22rz5.

Literature Cited

1. Ser-Giacomi, E., Legrand, T., Hernandez-Carrasco, I. & Rossi, V. Explicit and implicit network connectivity: Analytical formulation and application to transport processes. *Phys. Rev. E* 103, 042309 (2021).
2. Marzouk, Z., Aurelle, D., Said, K. & Chenuil, A. Cryptic lineages and high population genetic structure in the exploited marine snail *Hexaplex trunculus* (Gastropoda: Muricidae). *Biol. J. Linn. Soc.* 122, 411–428 (2017).
3. Rousset, F. Inferences from spatial population genetics. in *Handbook of statistical genetics* vol. 4 23 (2001).
4. Crandall, E. D., Tremblay, E. A. & Barber, P. H. Coalescent and biophysical models of stepping-stone gene flow in neritid snails. *Mol. Ecol.* 21, 5579–5598 (2012).
5. Kool, J. T., Paris, C. B., Andréfouët, S. & Cowen, R. K. Complex migration and the development of genetic structure in subdivided populations: an example from Caribbean coral reef ecosystems. *Ecography* 33, 597–606 (2010).
6. Jahnke, M. et al. Seascape genetics and biophysical connectivity modelling support conservation of the seagrass *Zostera marina* in the Skagerrak–Kattegat region of the eastern North Sea. *Evol. Appl.* 11, 645–661 (2018).
7. Guillot, G. & Rousset, F. Dismantling the Mantel tests. *Methods Ecol. Evol.* 4, 336–344 (2013).
8. Selkoe, K. A. et al. A decade of seascape genetics: contributions to basic and applied marine connectivity. *Mar. Ecol. Prog. Ser.* 554, 1–19 (2016).
9. Wang, I. J., Glor, R. E. & Losos, J. B. Quantifying the roles of ecology and geography in spatial genetic divergence. *Ecol. Lett.* 16, 175–182 (2013).
10. Legendre, P. & Fortin, M.-J. Comparison of the Mantel test and alternative approaches for detecting complex multivariate relationships in the spatial analysis of genetic data. *Mol. Ecol. Resour.* 10, 831–844 (2010).
11. Legendre, P., Fortin, M.-J. & Borcard, D. Should the Mantel test be used in spatial analysis? *Methods Ecol. Evol.* 6, 1239–1247 (2015).
12. Boulanger, E., Dalongeville, A., Andrello, M., Mouillot, D. & Manel, S. Spatial graphs highlight how multi-generational dispersal shapes landscape genetic patterns. *Ecography* 43, 1167–1179 (2020).
13. Jahnke, M. & Jonsson, P. R. Biophysical models of dispersal contribute to seascape genetic analyses. *Philos. Trans. R. Soc. B Biol. Sci.* 377, 20210024 (2022).
14. Jenkins, D. G. et al. A meta-analysis of isolation by distance: relic or reference standard for landscape genetics? *Ecography* 33, 315–320 (2010).
15. McRae, B. H. & Beier, P. Circuit theory predicts gene flow in plant and animal populations. *Proc. Natl. Acad. Sci.* 104, 19885–19890 (2007).

REVIEWERS' COMMENTS

Reviewer #1 (Remarks to the Author):

The authors have done a good job in the revised version of the manuscript and responding all the reviewers' suggestions. The methods are clear and the results seem sound and well supported by the new analyses. The manuscript is well written, and represents an interesting contribution to using multi-generation dispersal models to assess connectivity.

Minor comment

Table SI-12 should be Table SI-10

Reviewer #2 (Remarks to the Author):

The authors have done a good job at addressing the reviewer's comments and I have only minor remaining comments at this point.

One general issue is that on several occasions the authors provided a response to the reviewers in the response letter but it is not clear whether and if so where in the manuscript this has been addressed. The author's responses are generally sound but if they are not reflected in the ms this will not help readers who may have the same question.

In these lines would suggest to better justify and explain the use of $F_{st}/(1-F_{st})$ in the ms. Also given that $F_{st}/(1-F_{st})$ was considered instead of F_{st} would it make sense to present $F_{st}/(1-F_{st})$ instead of F_{st} in Figures 4b and 4c?

Minor suggestions

Line 45: whom or whose?

Line 66: counteractS?

Line 81: which or whose?

Line 229-230: "sampling spatial coverage" is a bit vague and potentially confusing, explain here what Db_{tw} exactly refers to since this is the first occurrence in the ms I believe.

Line 275: consider rewording "displayed significant prediction"

REVIEWERS' COMMENTS

Reviewer #1 (Remarks to the Author):

The authors have done a good job in the revised version of the manuscript and responding all the reviewers' suggestions. The methods are clear and the results seem sound and well supported by the new analyses. The manuscript is well written, and represents an interesting contribution to using multi-generation dispersal models to assess connectivity.

Thank you very much for the positive comments.

Minor comment

Table SI-12 should be Table SI-10

Thanks, it has been modified in the revised manuscript.

Reviewer #2 (Remarks to the Author):

The authors have done a good job at addressing the reviewer's comments and I have only minor remaining comments at this point. One general issue is that on several occasions the authors provided a response to the reviewers in the response letter but it is not clear whether and if so where in the manuscript this has been addressed. The author's responses are generally sound but if they are not reflected in the ms this will not help readers who may have the same question.

Thank you for the positive comments. We would like to stress that, despite the doubts, all relevant comments from previous rounds were indeed properly addressed in the manuscript.

In these lines would suggest to better justify and explain the use of $F_{st}/(1-F_{st})$ in the ms. Also given that $F_{st}/(1-F_{st})$ was considered instead of F_{st} would it make sense to present $F_{st}/(1-F_{st})$ instead of F_{st} in Figures 4b and 4c?

We agree with the reviewer on this point: we add one sentence to better explain the use of $F_{st}/(1-F_{st})$ in the ms (l. 179). Moreover, it makes totally sense to present $F_{st}/(1-F_{st})$ instead of F_{st} in Figures 4b and 4c. As such, the panels b and c have been changed accordingly.

Minor suggestions

Line 45: whom or whose?

"Whose" refers to a possessive in relative clause and indicate who or what something belongs to.

"Whom" is used in relative clauses to replace an object of a verb. As such, "whom" is correctly used in the sentence "For species *pronoun* migration is realized through dispersal driven by atmospheric or oceanic flows".

Line 66: counteractS?

Thanks, it has been modified in the revised manuscript.

Line 81: which or whose?

“Whose” conveys a very strong notion of possession, that “which” does not. In the sentence “This study focuses on sessile populations, *relative pronoun* adults have no or little displacement abilities”, the *relative pronoun* does not have a notion of possession. As such, which is well adapted in this sentence.

Line 229-230: "sampling spatial coverage" is a bit vague and potentially confusing, explain here what Dbtw exactly refers to since this is the first occurrence in the ms I believe.

Thanks, it has been detailed in the revised manuscript.

Line 275: consider rewording "displayed significant prediction"

Thanks, the corresponding sentence has been rewritten in the revised manuscript.